# Bayesian Meta-Learning with Expert Feedback for Task-Shift Adaptation through Causal Embeddings

**Lotta Mäkinen** [1 2]   **Jorge Loría** [1 2]   **Samuel Kaski** [1 2 3]

## Abstract

Meta-learning methods perform well on new within-distribution tasks but often fail when adapting to out-of-distribution target tasks, where transfer from source tasks can induce negative transfer. We propose a causally-aware Bayesian meta-learning method, by conditioning task-specific priors on precomputed latent causal task embeddings, enabling transfer based on mechanistic similarity rather than spurious correlations. Our approach explicitly considers realistic deployment settings where access to target-task data is limited, and adaptation relies on noisy (expert-provided) pairwise judgments of causal similarity between source and target tasks. We provide a theoretical analysis showing that conditioning on causal embeddings controls prior mismatch and mitigates negative transfer under task shift. Empirically, we demonstrate reductions in negative transfer and improved out-of-distribution adaptation in controlled simulations and a real-world clinical prediction setting for cross-disease transfer, where causal embeddings align with underlying clinical mechanisms; we include the judgments from a medical expert in the clinical prediction task and obtain improved performance in predictions of unseen diseases.

## 1. Introduction

A fundamental challenge in transfer learning and meta-learning is adapting to new tasks whose data-generating mechanisms differ from those encountered during training. In meta-learning, each task has its own conditional distri-

bution, and **task-level** distribution shift occurs when the target task is generated by a different underlying mechanism from the source tasks. This shift can severely degrade predictive performance (Quiñonero-Candela et al., 2022), making this issue critical in many domains such as healthcare. Clinically similar manifestations across patients and diseases may arise from distinct biological mechanisms (Subbaswamy & Saria, 2020): for example, type 1 and 2 diabetes share overlapping symptoms but arise from different mechanisms, namely impaired insulin *production* versus impaired insulin *response*. Hence, models trained across tasks may transfer spurious correlations rather than causal mechanisms, leading to worse performance when deployed on new tasks (Wang et al., 2019).

On the other hand, under appropriate assumptions, **causal relationships** remain invariant across environments and are therefore stable under distribution shift (Arjovsky et al., 2019; Pearl, 2009). If the causal mechanism of each task were known, identifying which source tasks are relevant for a new target task would be straightforward, since tasks with similar mechanisms would be expected to generalize well to one another. In practice, however, the causal mechanisms are **not** known, and existing causal discovery and inference methods typically require shared feature spaces or joint access to data across tasks (Peters et al., 2016; Lorch et al., 2021). The problem is that when source and target tasks originate from different datasets, it is *impossible* to assess the causal relationships between source and target tasks from observational data alone. This situation is prevalent in healthcare settings, for example, in rare and emerging diseases where recorded outcomes are too scarce to estimate causal effects, despite expert knowledge of the underlying mechanisms being available. The prevalence of this issue compounds due to privacy concerns and strict laws governing electronic health records, where some datasets might not contain information of diseases that are available in other datasets.

Reasoning about causal relationships is a core part of clinical practice. Through differential diagnosis, clinicians routinely compare new patient cases to previously seen ones, reasoning about which underlying mechanisms best explain the observed presentation (Eva, 2005; Pelaccia et al., 2011).

---

[1]ELLIS Institute Finland, Espoo, Finland [2]Department of Computer Science, Aalto University, Espoo, Finland [3]Department of Computer Science, University of Manchester, United Kingdom. Correspondence to: Lotta Mäkinen <lotta.makinen@aalto.fi>, Jorge Loría <jorge.loria@aalto.fi>.

*Proceedings of the 43rd International Conference on Machine Learning*, Seoul, South Korea. PMLR 306, 2026. Copyright 2026 by the author(s).

Such expert knowledge provides exactly the causal information that is *unidentifiable* from data alone. Despite this, no prior work has leveraged expert judgments about causal similarity to align target and source tasks.

Although **meta-learning** is a powerful framework for fast adaptation to new tasks using only a few samples, it still struggles when faced with out-of-distribution tasks (Hospedales et al., 2021). Most meta-learning approaches optimize a single shared initialization or prior across source tasks, which is then used to adapt to a new target task. As a result, the meta-learner cannot distinguish the source tasks that are most relevant for a given target task, and the shared prior may bias adaptation in a wrong direction when the target task lies outside the source-task distribution, leading to negative transfer. Meta-learning methods that incorporate task similarity, rely on "closeness" in the model parameters or learned features, without capturing causal structures (Yao et al., 2019; Zhou et al., 2021).

Our approach leverages causal task structures to guide transfer under task-level distribution shift. During meta-training, each source task is embedded into a latent causal task embedding space designed to capture stable mechanistic relationships between tasks. These causal task embeddings parameterize the task-specific prior of a Bayesian meta-learner, modulating transfer by the causal relationships between tasks rather than (possibly spurious!) correlations.

At deployment, when the embedding of a new target task is unknown, we infer its position in the causal embedding space using pairwise expert task similarity judgments between the target and source tasks. We assume access to a domain expert that provides (noisy) pairwise judgments about the relative causal similarity between tasks, without access to task observations. The inferred target embedding then defines a task-adaptive prior that emphasizes causally aligned source tasks, mitigating negative transfer under distribution shift. We demonstrate the effectiveness of this approach through theoretical analysis and experiments on both synthetic benchmarks and in a real-world clinical cross-disease prediction setting.

Our main contributions are:

- A causally-aware Bayesian meta-learning method that conditions task-specific priors on a causal task embedding space and uses a Bayesian expert preference model over pairwise similarity queries to infer the target-task embedding without requiring access to target-task data, described in Sections 4 and 5.

- A theoretical analysis characterizing negative transfer in terms of prior mismatch under task shift, presented in Section 6.

- Numerical experiments on large-scale medical data showing that our approach improves adaptation and

reduces negative transfer under distribution shift, compared to alternative methods which do not incorporate expert knowledge, shown in Section 7.

## 2. Preliminaries

### 2.1. Bayesian Meta-Learning

We build on the amortized hierarchical Bayesian meta-learning formulation (Ravi & Beatson, 2019), which we summarize in this section. The generative model assumes shared global parameters of a neural network and task-specific parameters in the same space. The global parameters are denoted by $\theta \in \mathbb{R}^p$, and the task-specific parameters by $\phi_t \in \mathbb{R}^p$, for $t \in \mathcal{T}$; following the hierarchical structure

$$\theta \sim p(\theta), \tag{1}$$

$$\phi_t \mid \theta \overset{i.i.d.}{\sim} \mathcal{N}(\theta, \sigma^2 I_p), \text{ for } t \in \mathcal{T}, \text{ and} \tag{2}$$

$$\mathbf{y}_{t,i} \mid \mathbf{x}_{t,i}, \phi_t \overset{indep.}{\sim} p(\mathbf{y}_{t,i} \mid \phi_t, \mathbf{x}_{t,i}), \tag{3}$$

for $i = 1, \ldots, M_t$. The task-prior $\phi_t \mid \theta$ ensures that the task-specific parameters remain close to the global parameters $\theta$. In this paper we follow the common choice of independent Gaussian random variables, but this is easily adapted to other general priors. The likelihood for task $t$ is denoted by $p(\mathbf{y} \mid \phi_t, \mathbf{x})$, with parameters encoded by $\phi_t$ (e.g., the mean and variance are functions of $\phi_t$, and $\mathbf{x}$). Each task dataset is given by $\mathcal{D}_t = \{(\mathbf{x}_{t,i}, \mathbf{y}_{t,i})\}_{i=1}^{M_t}$ and is further divided into a support set $\mathcal{D}_t^{(s)}$ and a query set $\mathcal{D}_t^{(q)}$, which simulates train-test episodes inside the meta-training. Amortized variational inference is used, following Ravi & Beatson (2019), to learn a global variational distribution $q_\lambda(\theta)$ and task-level variational distributions $q_{\psi_t}(\phi_t \mid \mathcal{D}_t^{(s)})$, where the $\lambda$ are the variational global parameters and the $\psi_t$ the variational task-specific parameters of the task posterior.

Typical approaches follow bi-level optimization objective with nested inner and outer loss functions. The inner task-level loss function is

$$\mathcal{L}_1(\psi_t, \mathcal{D}_t^{(s)}) := -\mathbb{E}_{q_{\psi_t}(\phi_t \mid \mathcal{D}_t^{(s)})} \Big[ \log p(\mathbf{y}_t^{(s)} \mid \mathbf{x}_t^{(s)}, \phi_t) \Big]$$
$$+ D_{\mathrm{KL}} \Big( q_{\psi_t}(\phi_t \mid \mathcal{D}_t^{(s)}) \parallel p(\phi_t \mid \theta) \Big), \tag{4}$$

with outer global loss function given by

$$\mathcal{L}_2(\psi_t, \mathcal{D}_t^{(q)}, \lambda) := -\mathbb{E}_{q_{\psi_t}(\phi_t \mid \mathcal{D}_t^{(s)})} \Big[ \log p(\mathbf{y}_t^{(q)} \mid \mathbf{x}_t^{(q)}, \phi_t) \Big]$$
$$+ D_{\mathrm{KL}} \Big( q_\lambda(\theta) \parallel p(\theta) \Big). \tag{5}$$

The bi-level problem has an outer optimization of

$$\min_\lambda \mathbb{E}_{t \sim p(\mathcal{T})} \Big[ \mathcal{L}_2 \left( \psi_t^*, \mathcal{D}_t^{(q)}, \lambda \right) \Big], \tag{6}$$

where $\psi_t^* = \arg\min_{\psi_t} \mathcal{L}_1(\psi_t, \mathcal{D}_t^{(s)})$ is a solution of the inner optimization.

During meta-training a shared prior $p(\phi_t \mid \theta)$ across tasks is used to learn a posterior distribution of the parameters for each task, and at meta-test time, the model is presented with a new target task $t'$. The task-specific parameters are adapted by minimizing $\mathcal{L}_1(\psi_{t'}, \mathcal{D}_{t'}^{(s)})$ using the shared prior. Predictions are made on held-out test data. A key limitation of this formulation is that when tasks differ in their data-generating mechanisms, enforcing a single global prior for all possible target tasks can lead to worse performance on out-of-distribution target tasks, leading to negative transfer (Wang et al., 2019). In Section 4 we extend this framework by conditioning the prior $p(\phi_t \mid \theta)$ on causal task embeddings, allowing the prior to vary across tasks. Furthermore, in Section 6 we explore how this affects transfer in traditional methods and how our approach is better.

## 2.2. Structural Causal Models

We follow the structural causal model (SCM) notation (Pearl, 2009). A task-specific SCM is a tuple $\mathcal{W} = (\mathbf{V}, \mathbf{U}, F, P_U)$, where $\mathbf{V} = \{X_1, \ldots, X_d\}$ are the endogenous variables, $\mathbf{U} = \{U_1, \ldots, U_d\}$ are exogenous noise variables, $F = \{f_1, \ldots, f_d\}$ is the set of structural equations $X_i = f_i(\mathrm{Pa}(X_i), U_i)$, with $\mathrm{Pa}(X_i) \subseteq \mathbf{V}$ denoting the causal parents of $X_i$, and $P_U$ is the joint distribution over the noise variables. An SCM induces a directed acyclic graph (DAG) where each variable in $\mathbf{V}$ is a vertex and edges encode parent-child relations defined by $\mathrm{Pa}(X_i)$.

## 3. Related Work

**Meta-learning under task-level distribution shift.** Meta-learning provides a framework for transferring inductive bias across multiple tasks by learning shared model parameters or priors. Existing approaches include gradient-based methods, such as model agnostic meta-learning (Finn et al., 2017), and Bayesian meta-learning methods, which model tasks through a shared global prior over task-specific parameters and offer a principled way to reason about uncertainty (Grant et al., 2018; Finn et al., 2018; Yoon et al., 2018; Ravi & Beatson, 2019). However, most meta-learning methods implicitly assume that source and target tasks are identically and independently drawn from a common task distribution. When this assumption is violated, the learned inductive bias becomes misspecified and transfer can even lead to negative transfer (Wang et al., 2019). Recent theoretical works formalize this by showing that meta-generalization error depends explicitly on a divergence between source and target task distributions: implying that performance degrades when target tasks move farther from source tasks (Fallah et al., 2021; Jose & Simeone, 2021; Chen et al., 2021). In order to mitigate heterogeneity among tasks, several approaches cluster source tasks or learn multiple task-specific priors or initializations (Zhou et al., 2021; Yao et al.,

2019). Other works embed tasks into latent spaces derived from data and compute task similarities to guide transfer (Achille et al., 2019; Chen et al., 2020). However, these approaches define the task similarity using observed features or model parameters, which are unstable under distribution shift. More recent work has explored modeling similarity between source tasks at the level of causal mechanisms to group tasks during training to improve generalization within the same task distribution (Wharrie et al., 2024). Crucially, none of these methods address how to align target and source tasks and infer the causal relationships between them, particularly when no target task data is available.

**Causal inference and task heterogeneity.** From a causal perspective, task heterogeneity arises from differences in the underlying data-generating processes (Pearl, 2009). If the task-specific structural causal models (SCMs) are fully known, tasks could be compared directly by differences in their directed acyclic graphs (DAGs) or structural equations. Recent Bayesian causal discovery methods, such as DiBS (Lorch et al., 2021), aim to infer full posterior distributions over causal graph structures from data. While powerful, inferring complete SCMs is often impractical in real-world settings, particularly in high-dimensional problems or in the presence of latent confounding. Instead of full causal discovery, many causal inference (CI) approaches estimate task-specific causal effects. Methods such as instrumental variable estimation (Angrist et al., 1996) and invariant causal prediction (Peters et al., 2016) produce estimates of the direction and magnitude of causal relationships, which can be summarized as causal representation vectors (Schölkopf et al., 2021). These representations can serve as proxies for the underlying causal mechanisms, enabling task comparison even when full SCM discovery is impossible. However, existing CI methods require access to task-level observational data, and cannot be applied to assess similarity between disjoint source and target tasks using data alone.

**Expert knowledge in learning systems.** Incorporating expert knowledge into learning systems has been studied in a variety of settings, particularly through preference learning and pairwise comparison methods. Ranking-based models are a classical way of eliciting relative information from experts, formalized by Thurstone (1927) and Bradley & Terry (1952). Preference learning extends these models by placing a probabilistic model over latent utilities or similarities and modeling expert feedback as noisy but consistent relative judgments (Chu & Ghahramani, 2005). These models have been naturally connected to active learning for selecting the most informative queries (Houlsby et al., 2011). Expert knowledge has also been incorporated into CI, particularly in the context of learning causal structures. Prior work has explored learning causal Bayesian networks with expert elicitation (Constantinou et al., 2016). Recently, Björkman

et al. (2026) proposed a Bayesian experimental design approach to elicit expert knowledge for discovering mixtures of DAGs in heterogeneous domains. However, these approaches identify causal structure *within* a dataset, rather than aligning a new task to previously learned tasks when target data is unavailable. In meta-learning, incorporating expert knowledge remains an open challenge highlighted by recent reviews (Vettoruzzo et al., 2024). In particular, existing meta-learning methods do not leverage expert feedback to align target tasks with source tasks at adaptation time. In contrast, our approach uses expert pairwise similarity judgments to align a (previously unseen) task with source tasks based on the causal mechanisms, enabling transfer under task-level shifts.

# 4. Incorporating Causal Embeddings to a Bayesian Meta-Learning Model

## 4.1. Causal Embeddings

Assume access to latent *causal embeddings* $z \in \mathcal{Z} \subseteq \mathbb{R}^d$, equipped with a metric $d_{\mathcal{Z}} : \mathcal{Z} \times \mathcal{Z} \to \mathbb{R}_{\geq 0}$. We use the Euclidean metric for simplicity, although our framework is agnostic to the choice of metric.

**Definition 1.** For each task $t \in \mathcal{T}$, we define a causal task embedding $z_t \in \mathcal{Z}$, a low-dimensional vector representation summarizing the causal mechanistic structure of the task.

The embedding $z_t$ is constructed by mapping the vector representation of the underlying SCM. To construct said vector representation, we use invariant causal prediction (ICP; Peters et al., 2016) and an instrumental variables approach with Mendelian randomization (MR; Didelez & Sheehan, 2007), both described in App. E.

Task embeddings are constructed to summarize causal mechanisms. As such, distances in $\mathcal{Z}$ can be interpreted as causal relatedness. We introduce the following notion of proximity in the embedding space.

**Definition 2.** Two tasks $t$ and $t'$ are $\varepsilon$-similar if $d_{\mathcal{Z}}(z_t, z_{t'}) \leq \varepsilon$, where $d_{\mathcal{Z}}$ is the metric on $\mathcal{Z}$.

For a specific task $t \in \mathcal{T}$, the true outcome generating distribution is denoted by $p_t(y \mid \mathbf{x})$. The following key assumption helps link causal structure to predictive behavior.

**Assumption 3.** Given two $\varepsilon$-similar tasks $t, t'$, their predictive conditional distributions $p_t(y \mid \mathbf{x})$ and $p_{t'}(y \mid \mathbf{x})$ are at a distance of $\delta_{\varepsilon}$.

Assumption 3 is a smoothness condition on $p_t(y \mid \mathbf{x})$ with respect to the latent embeddings $z_t$. Formalizing the idea that similarity in the underlying causal mechanisms can be translated into similarity in the predictive behavior. This assumption is empirically validated in App. E.4.

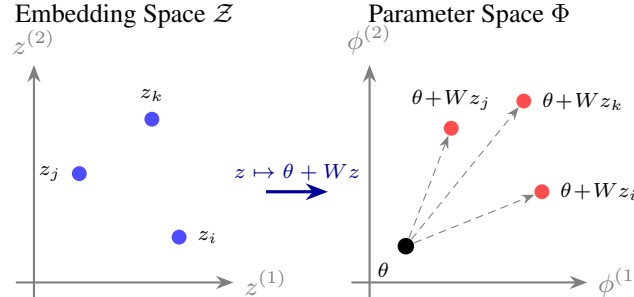

*Figure 1.* Illustration of the causal embedding space (*left*) and the mapping to the parameter space (*right*). Each task $t$ has an embedding $z_t$, encoding its causal structure. The linear map $z \mapsto \theta + Wz$ transforms embeddings into task-specific priors, enabling tasks with similar causal mechanisms to have similar priors.

In the rest of the article, we use the task embeddings $z_t$ to condition the task-specific prior, such that the geometry of $\mathcal{Z}$ implicitly modulates information transfer across tasks.

## 4.2. Causally-Aware Bayesian Meta-Learning

We extend the Bayesian meta-learning framework from Section 2.1 by conditioning the task-specific prior on causal embeddings. Eq. (2), the standard task-specific prior, centers all tasks around the same global parameter $\theta$. We modify it to the embedding-aware prior

$$\phi_t \mid z_t, \theta \stackrel{indep.}{\sim} \mathcal{N}(\theta + Wz_t, \sigma^2 I_p), \text{ for } t \in \mathcal{T}, \quad (7)$$

where $W \in \mathbb{R}^{p \times d}$ is a learnable weight matrix mapping embeddings to parameter space, and $\sigma^2$ is a prior variance. The prior mean $\theta + Wz_t$ is task-specific through the embedding $z_t$, while the prior variance is shared across all tasks.

This linear parameterization admits a geometric interpretation (Figure 1). The global parameter $\theta$ is a starting point in the parameter space $\Phi$, and each task embedding $z_t$ induces a displacement $Wz_t$ within the affine subspace $\{\theta + Wz : z \in \mathcal{Z}\}$, arriving at a task-specific prior mean at $\theta + Wz_t$. Tasks that lie close in the embedding space $\mathcal{Z}$ induce similar prior means; this aligns with Assumption 3. When $z_t = 0$ or $W = 0$, the prior reduces to $\mathcal{N}(\theta, \sigma^2 I)$, the standard meta-learning prior.

Conditioning the prior mean allows task-specific inductive biases to vary smoothly across tasks as a function of their causal similarity, influencing the direction in which the model adapts during task-specific learning. In contrast, keeping the prior variance global isolates the effect of task similarity on the prior mean, promotes stable adaptation across tasks, and simplifies the theoretical analysis of generalization under task shift. Although more expressive mappings could be considered (e.g., neural networks), linear mappings yield favorable theoretical properties and enable analysis of the transfer between task similarity while remaining suf-

ficient to capture task-level structures in our setting. We empirically validate this choice in App. C.3 with synthetic data and in App. D.6 with real-world data, showing that a nonlinear variant underperforms the linear parametrization.

We instantiate the proposed model within the standard amortized Bayesian meta-learning framework, described in Section 2.1. Consider a dataset $\mathcal{D}_t = \{(\mathbf{x}_{t,i}, y_{t,i})\}_{i=1}^{M_t}$ with $M_t$ samples for task $t$ drawn from this distribution, and assume that the tasks are separated into source and target, denoted by $\mathcal{T} = \mathcal{T}_{\text{source}} \cup \mathcal{T}_{\text{target}}$. During meta-training, the model observes a set of source tasks $\mathcal{T}_{\text{source}}$ sampled from a task distribution $t \sim p(\mathcal{T}_{\text{source}})$. For each source task $t$, we assume a causal task embedding $z_t$ is available, and use it to construct an embedding-conditioned prior $p(\phi_t \mid z_t, \theta)$. Task-specific parameters are adapted on support data $\mathcal{D}_t^{(s)}$ by minimizing the inner variational objective

$$
\mathcal{L}_1(\psi_t, \mathcal{D}_t^{(s)}, z_t) := -\mathbb{E}_{q_{\psi_t}(\phi_t \mid \mathcal{D}_t^{(s)})} \Big[ \log p(y_t^{(s)} \mid \mathbf{x}_t^{(s)}, \phi_t) \Big]
$$
$$
+ D_{\text{KL}}\Big( q_{\psi_t}(\phi_t \mid \mathcal{D}_t^{(s)}) \,\|\, p(\phi_t \mid z_t, \theta) \Big),
$$

instead of Eq. (4). Conditioning the prior on $z_t$ directly influences the KL regularization term in the inner objective ensuring the posterior does not drift too far from a causally aligned prior. The task predictor is parameterized by a Bayesian neural network, where the task parameters $\phi_t$ correspond to the weights of a task prediction head. Full algorithms and optimization details are provided in App. B.

At meta-test time, the model is presented with a previously unseen target task $t'$ sampled from its task distribution $t' \sim q(\mathcal{T}_{\text{target}})$. If $p(\mathcal{T}_{\text{source}}) \neq q(\mathcal{T}_{\text{target}})$, task-level distribution-shift occurs and the target task distribution will not be aligned with the source task distribution. The target task embedding $z_{t'}$ is used to construct a task-specific prior $p(\phi_{t'} \mid z_{t'}, \theta)$, which guides adaptation to the target task using its observed support data. For adaptation and prediction in new tasks we use Alg. B.2.

## 5. Expert-Guided Inference of Target Task Embeddings

In a realistic deployment scenario, the source and target datasets are not simultaneously available. Meaning that the source and target tasks ($t$ and $t'$, respectively) cannot be easily embedded into $\mathcal{Z}$ at training time. We assume that it is possible to embed the source tasks into $\mathcal{Z}$, and that the target task embedding $z_{t'}$ is unknown at deployment time, which is key to make predictions in our model.

To address this challenge, we propose inferring the target task embedding using structured feedback from a *domain expert*. The key assumption is that while the expert does not have access to the underlying data, they have domain knowledge of the underlying causal mechanisms. As such,

they can indicate which of two tasks is closer to the new task. For example, a doctor can usually assess the relative similarity between two diseases or two patients based on their medical expertise, since that is the reasoning typically used in their daily work. We elicit expert knowledge through pairwise similarity comparisons between source tasks and the target task. Then we use these comparisons to infer the target task embedding $z_{t'}$ by aligning it to the embeddings of the source tasks.

We formalize expert feedback using comparison queries in pairs. Queries are denoted by $\xi_b = (i_b, j_b)$, for $b \in \{1, \ldots, B\}$, where $i_b, j_b \in \mathcal{T}_{\text{source}}$ are two source tasks with known embeddings. At each iteration $b$, the expert is asked:

> Is source task $i_b$ more similar to target task $t'$ than source task $j_b$?

The expert response is denoted by $c_b \in \{0, 1\}$, where $c_n = 1$ indicates that task $i_b$ is judged to be closer to the target task than $j_b$. Collecting $B$ such queries yields a dataset of expert feedback $\mathcal{C} = \{(\xi_b, c_b)\}_{b=1}^B$.

Let $z_{t'} \in \mathbb{R}^d$ denote the latent target task embedding. For a query $\xi = (i, j)$, we define the relative dissimilarity between the source tasks with respect to the target task as

$$
\Delta(\xi; z_{t'}) = d_{\mathcal{Z}}(z_{t'}, z_j) - d_{\mathcal{Z}}(z_{t'}, z_i)
$$
$$
= \|z_{t'} - z_j\|_2 - \|z_{t'} - z_i\|_2.
$$

The quantity $\Delta(\xi; z_{t'})$ is a deterministic function of the known embeddings ($z_i$ and $z_j$) and encodes their relative geometry in the task-embedding space. A positive value of $\Delta(\xi; z_{t'})$ indicates that source task $i$ is closer to the target task $t'$ than $j$ is to $t'$.

Although the value of $\Delta(\xi; z_{t'})$ is deterministic (given the $z$s) the expert could *potentially* give incorrect responses and as such we assign a likelihood to the responses. To this end, the expert responses are modeled by a probit likelihood; a standard approach in preference learning (Chu & Ghahramani, 2005). The likelihood for each comparison is

$$
p(c = 1 \mid z_{t'}, \xi, \tau) = \Phi(\tau \, \Delta(\xi; z_{t'})), \qquad (8)
$$

where $\Phi(\cdot)$ denotes the standard normal cumulative distribution function and $\tau > 0$ controls the noise level in expert judgments. We fix $\tau = 1$ during inference (to avoid identifiability issues) and assign a Gaussian prior on the target-task embedding: $z_{t'} \sim \mathcal{N}(0, I_d)$.

Given a set of expert responses $\mathcal{C}$, the posterior is

$$
p(z_{t'} \mid \mathcal{C}) \propto p(z_{t'}) \prod_{n=1}^N p(c_n \mid z_{t'}, \xi_n, \tau),
$$

which we approximate using stochastic variational inference with $q_\varphi(z_{t'}) = \mathcal{N}\left(\mu_q, \sigma_q^2 I_d\right)$, as the approximating family.

To select the queries that will be presented to the expert we use Bayesian active learning by disagreement (BALD; Houlsby et al., 2011). At each iteration, we select the query $\xi = (i, j)$ that maximizes the expected information gain (EIG) about the target embedding $z_{t'}$:

$$\xi^\star = \arg\max_\xi \ \text{EIG}(\xi), \ \text{EIG}(\xi) = I(y_\xi; z_{t'} \mid \mathcal{C}), \quad (9)$$

where $I(\cdot; \cdot)$ denotes the mutual information and $y_\xi$ is the (future) expert response to query $\xi$. We use the objective in BALD (Houlsby et al., 2011), with the decomposition

$$\text{EIG}(\xi) = H[y_\xi] - \mathbb{E}_{z_{t'} \sim p(z_{t'} \mid \mathcal{C})}[H[y_\xi \mid z_{t'}]],$$

where $H[p] = -p \log p - (1 - p) \log(1 - p)$ is the binary entropy function. Both entropy terms are approximated by sampling from the current variational posterior $q_\varphi(z_{t'})$. The complete active learning algorithm is in App. B.2.

Finally, in Algorithm 1 we present the full meta-learning approach that incorporates expert knowledge for prediction in new tasks via causal embeddings. This algorithm combines the meta-learning approach we developed in Section 4, with the expert knowledge and queries developed in this section, and adds the adaptation step. An implementation is freely available in www.github.com/lottamakinen/causal-meta-learning.

---

**Algorithm 1** Causally-aware meta-learning with expert in the loop to predict in a new task

---

**Require:** source datasets $\mathcal{D} = \{\mathcal{D}_t\}_{t \in \mathcal{T}_{\text{source}}}$, target dataset $D_{t'}$, budget of expert queries $B$, expert comparisons $\mathcal{C}$
1: **Obtain** approximate posterior $q_\phi(z \mid \mathcal{D})$ using Alg. B.1
2: **Actively** learn $\hat{z}_{t'} = \mathbb{E}_q[z_{t'} \mid \mathcal{C}]$ with Alg. B.3
3: **Adapt** posterior to the target task $t'$, with Alg. B.2.

---

## 6. Error Decomposition and Negative Transfer Under Task-Level Shift

We analyze the adaptation to a target task $t' \sim q(\mathcal{T}_{\text{target}})$ under a task-level shift ($p(\mathcal{T}_{\text{source}}) \neq q(\mathcal{T}_{\text{target}})$). Our analysis focuses on (i) bounding the prior induced risk, which reflects how well the transferred inductive bias aligns with the target task, and (ii) the mitigation of negative learning it induces.

Denote by $\mathcal{R}_{t'}(\phi) = \mathbb{E}_{(x,y) \sim p_{t'}}[\ell(\phi; x, y)]$ the population risk on task $t'$. For the prior (Eq. (7)), the prior-induced risk is

$$\bar{\mathcal{R}}_t(z) = \mathbb{E}_{\phi \sim p(\phi \mid z)}[\mathcal{R}_t(\phi)].$$

Define the mismatch relative to the source-task-induced prior with embedding $\bar{z} := |\mathcal{T}_{\text{source}}|^{-1} \sum_{t \in \mathcal{T}_{\text{source}}} z_t$, as

$$\mathcal{E}_{\text{prior}}(t') := \left|\bar{\mathcal{R}}_{t'}(\hat{z}_{t'}) - \bar{\mathcal{R}}_{t'}(\bar{z})\right|.$$

Next, we formalize the intuition that nearby tasks in causal embedding space induce similar risks, and provides a decomposition of the generalization error in its three main components: the expert error, the causal discovery error, and the out-of-distribution error.

**Proposition 4.** *Assume the loss is bounded, $\ell(\phi; x, y) \in [0, M]$. If $z_1$ and $z_2$ are $\varepsilon$-similar, then for any task $t$,*

$$\left|\bar{\mathcal{R}}_t(z_1) - \bar{\mathcal{R}}_t(z_2)\right| \leq \frac{M\|W\|}{2\sigma}\varepsilon.$$

*For task $t'$, the mismatch satisfies*

$$\mathcal{E}_{\text{prior}}(t') \ \leq \ \frac{M\|W\|}{2\sigma}\left(\varepsilon_{\text{expert}} + \varepsilon_{\text{causal}} + \varepsilon_{\text{OOD}}\right), \quad (10)$$

*where $\|W\|$ is the spectral norm of the embedding weight matrix, $\varepsilon_{\text{OOD}} = \|z_{t'} - \bar{z}\|, \varepsilon_{\text{causal}} = \|\tilde{z}_{t'} - z_{t'}\|$, and $\varepsilon_{\text{expert}} = \|\hat{z}_{t'} - \tilde{z}_{t'}\|$; using $\tilde{z}_{t'}$ as the embedding recovered via causal discovery, and $\hat{z}_{t'}$ as the embedding used at deployment and inferred from expert feedback.*

The proof is given in App. A.1. Eq. (10) shows that prior mismatch grows at most *linearly* with geometric displacement in the embedding space. In particular, if the target task is $\varepsilon$-similar to the source tasks, then the induced prior mismatch is $O(\varepsilon)$. Furthermore, prior mismatch improves when (i) the target task is closer to the source-induced global prior (small $\varepsilon_{\text{OOD}}$), (ii) causal discovery is accurate (small $\varepsilon_{\text{causal}}$), and (iii) expert inference is accurate (small $\varepsilon_{\text{expert}}$). As such it provides three different ways to control the generalization error, instead of a single one.

Negative transfer occurs when an inductive bias from the source tasks induces a worse performance (higher risk) compared to training *without* transfer. For a method $X$ and task $t'$, negative transfer is

$$\text{NT}_X(t') := \mathcal{R}_{t'}(\hat{\phi}_{t'}^X) - \mathcal{R}_{t'}(\hat{\phi}_{t'}^{\text{NT}}), \quad (11)$$

where $\hat{\phi}_{t'}^{\text{NT}}$ is a predictor trained only on the target task.

Finally, we present Theorem 5, providing an interpretable condition to establish that negative transfer is **mitigated** by using a causal prior and a good-enough expert.

**Theorem 5.** *Under mild assumptions, stable adaptation, and a well-conditioned embedding map $W$, if*

$$\varepsilon_{\text{expert}} + \varepsilon_{\text{causal}} \ \leq \ C \cdot \varepsilon_{\text{OOD}},$$

*for a constant $C$. Then, the causal prior mitigates negative transfer relative to the global prior, i.e.,*

$$\text{NT}_{\text{causal}}(t') \leq \text{NT}_{\text{glob}}(t').$$

The proof and full statement are in App. A.2. The conditions of Theorem 5 require that the error from locating the target task in the causal embedding space is smaller than the level of task-level shift between the source and target tasks.

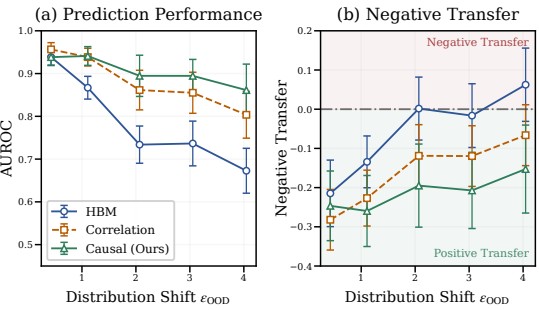

*Figure 2.* Performance of meta-learning models under increasing task-shift for Experiment 1. a) AUROC as a function of the distribution shift $\varepsilon_{\text{OOD}}$. b) Change in log loss relative to no transfer baseline (BNN), where negative values indicate positive transfer and positive values negative transfer. Error bars denote standard deviation across 30 runs.

# 7. Evaluation under Task-level Shifts

## 7.1. Synthetic Setting

We analyze the negative transfer, under distribution shift, of our method synthetic datasets. The data generation mechanisms are detailed in App. D.1.

**Experiment 1: Preventing Negative Transfer under Task Shift.** We evaluate whether conditioning the prior on causal task embeddings prevents negative transfer under task-level distribution shift in the presence of spurious correlations. For this, we compare three models: a global prior without task conditioning (meaning that it does not observe the $z_t$s), correlation-based task embeddings ($z_t$s inferred via correlations), and causal embeddings. Here we focus on the effect of OOD task-level shift. For this, we assume perfect causal discovery $\varepsilon_{\text{causal}} = 0$, and embeddings obtained with a perfect expert ($\varepsilon_{\text{expert}} = 0$). That is, the true $z_{t'}$ is known. In App. C.1 we show that the method stays robust under imperfect causal discovery when $\varepsilon_{\text{causal}} > 0$.

Figure 2a shows the AUROC for different levels of task shift ($\varepsilon_{\text{OOD}}$). The global prior and the correlation-based embeddings exhibit a degradation in predictive performance as the target task moves farther from the source distribution. In contrast, using the causal embeddings yields a more stable AUROC across all levels of task shift. Figure 2b illustrates this behavior through negative transfer (Eq. (11)), measured as the change in log-loss relative to a non-transfer baseline given by independently-trained task-specific Bayesian neural networks. Positive values indicate worse performance than the no-transfer baseline. The global prior has increasing negative transfer as $\varepsilon_{\text{OOD}}$ grows, reflecting a mismatch under larger task shifts. Correlation-based embeddings also exhibit larger negative transfer for larger shifts: highlighting the instability of spurious correlations. This instability is further shown by the increasing variance across runs, as in-

dicated by the widening error bars at higher shift levels. In contrast, the causal prior induces positive transfer across all OOD levels. The causal prior effectively mitigates negative transfer even under substantial task-level shift and spurious correlations, thereby explaining the stable performance.

Notably, correlation-based embeddings outperform the other approaches in the in-distribution case (i.e., $\varepsilon_{\text{OOD}} = 0.5$), where spurious correlations are predictive and therefore beneficial. However, this advantage disappears under task shift when these correlations no longer hold, while causal embeddings remain robust.

**Experiment 2: Using Expert Inferred Target Task Embeddings.** We next consider a more realistic deployment scenario in which the causal embedding of the target task is not known and must be inferred using expert feedback, as described in Section 5. We simulate an expert that provides noisy pairwise similarity judgments based on the true underlying causal task embeddings $z_t$. Expert responses are generated according to the likelihood in Eq. (8), with expert reliability controlled by the parameter $\tau_{\text{expert}}$. We set $\tau_{\text{expert}} = 2$, and report sensitivity analyses to expert noise in App. C.4. To reflect realistic imperfection in causal discovery, we further corrupt the source-task embeddings used by the meta-learner using directional perturbation with noise level $\sigma_c = 0.5$ (see App. C.1 for details). The pairwise queries presented to the expert are selected using BALD. App. C.5 shows comparisons to random query selection.

Figure 3a compares expert-inferred causal embeddings against meta-learning baselines under increasing task shift. The baselines included are a model agnostic meta-learning (MAML; Finn et al., 2017), a task-similarity aware meta-learning (TSA-MAML; Zhou et al., 2021), a deep kernel transfer model (DKT; Patacchiola et al., 2020), and an amortized hierarchical Bayesian meta-learning with a global prior (HBM; Ravi & Beatson, 2019). Both MAML and the global prior exhibit degraded performance as $\varepsilon_{\text{OOD}}$ increases. In contrast, conditioning the prior on expert-inferred causal embeddings yields performance close to the causal oracle embeddings without expert inference across all OOD levels. This indicates that expert feedback enables accurate recovery of a target task embedding that is well-aligned with the underlying causal mechanism, even when causal discovery is imperfect. DKT performs strongly across all shift levels in this synthetic setting. As a metric-based method utilizing Gaussian processes, DKT is well-suited to low-dimensional problems, and therefore serves as a strong baseline in this experiment. Similarly, TSA-MAML performs strongly in this synthetic setting, matching our method's performance, where the sufficient number of source tasks enables meaningful parameter-space clustering. As we show in Section 7.2, these advantages diminish in more complex real-world settings with fewer source tasks, where

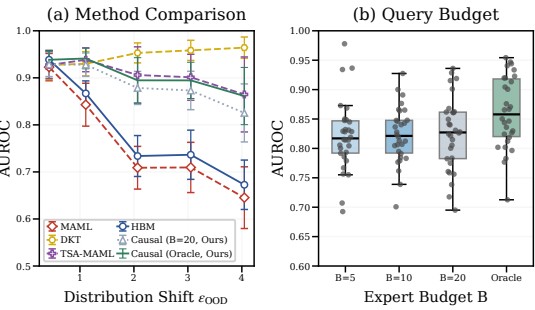

*Figure 3.* Method comparison with expert inferred embeddings for Experiment 2 across 30 runs. a) Average AUROC across increasing distribution shift $\varepsilon_{\text{OOD}}$ levels for our method against baselines, error bars denote SD. b) Effect of expert query budget on prediction performance, for a single task ($\varepsilon = 4.0$).

optimization-based and Bayesian meta-learning methods are more effective. Figure 3b shows the effect of expert query budget $B$ on performance for a single task ($\varepsilon_{\text{OOD}} = 4.0$). Increasing the number of expert queries consistently improves AUROC, reflecting progressively more accurate inference of the target task embedding.

### 7.2. Cross-Disease Meta-Learning in the UK Biobank

Next, we evaluate our proposed method on clinical prediction tasks derived from the UK Biobank (UKBB), a large-scale population cohort with longitudinal electronic health records. In contrast to the settings in Section 7.1, disease prediction tasks exhibit heterogeneous populations, complex pathways, and unknown confounders.

We define each task as a binary disease prediction problem, where the goal is to predict the presence of a certain disease after an index year using patient-level covariates, clinical history, and treatment endpoints. Tasks correspond to different disease endpoints, in the cardiovascular and respiratory systems. Details of the dataset are in App. D.5, along with a description of the diseases. This setting ensures distribution shift between the tasks, since diseases differ in prevalence, causal mechanisms, and risk factors.

Since the causal mechanisms are not observed, we construct embeddings using causal inference (CI) methods in the source tasks, using longitudinal data. We use three approaches: an instrumental variable approach with Mendelian Randomization (MR; Didelez & Sheehan, 2007); an invariant causal prediction (ICP; Peters et al., 2016); and a non-causal baseline, from observational temporal $\chi^2$-correlations (CHI2). See App. E for details of the embedding methods. The target task embeddings are inferred using a medical expert's feedback (first three rows of Table 1). We also simulate an expert whose responses are generated according to the likelihood (Eq. (8)), with $\tau_{\text{expert}} = 10$, labeled **n**. The simulated expert derives the task similarities

using the same CI method as used in the construction of the source task embeddings, applied to a larger reference dataset that includes source tasks, target tasks, and also additional related disease endpoints. This reflects their broader domain knowledge. We also consider the data-informed setting, in which the meta-learner directly uses these embeddings from a reference dataset, bypassing the expert model, and providing an upper bound on data-informed performance.

We compare our approach against the same meta-learning baselines in Section 7.1. The performance of our model is assessed with the three embeddings (CHI2, ICP, MR) considered, using both the expert-inferred embeddings and oracle embeddings. App. D.5 includes additional details for all methods. Table 1 displays the average AUROC (10 seeds) of each method in four out-of-distribution tasks. Across tasks, standard meta-learning baselines (MAML, TSA-MAML, DKT, HBM) generally improve over the "no-transfer" baseline indicating that task transfer is beneficial, although with modest and task-dependent gains. Our causal embedding-conditioned method further improves performance.

For the embeddings inferred via the real medical expert, ICP yields performance close to the oracle, with consistent improvements across all target tasks. For the simulated expert, we observe improvements on tasks J44 and J45 when using ICP embeddings. In contrast, MR and CHI2 exhibit less consistent behavior across tasks, reflecting differences in the task similarities induced by each method. As shown in App. E.3, ICP embeddings induce a more structured embedding space, and identify J44 and J45 as the most OOD tasks relative to the source-task mean, which explains the stronger performance gains observed for these tasks. Additional results are provided in App. D.6.

*Table 1.* Average AUROC (SD) on target tasks from the UKBB, over 10 seeds. Our method using causal embeddings (ICP, MR) and an expert (**m**edical expert, **n**oisy expert from data, or **d**ata without noise) perform better than the baselines. CHI2 corresponds to our method using correlations. Best in **bold**.

| Method | J44 | J45 | G45 | I21 |
|---|---|---|---|---|
| ICP, **m**. | **0.823 (.005)** | 0.620 (.006) | **0.676 (.009)** | 0.711 (.006) |
| MR, **m**. | 0.778 (.008) | 0.601 (.006) | 0.665 (.006) | 0.711 (.008) |
| CHI2, **m**. | 0.740 (.033) | 0.560 (.022) | 0.655 (.016) | 0.704 (.015) |
| ICP, **n**. | 0.820 (.007) | 0.620 (.007) | 0.655 (.007) | 0.693 (.008) |
| MR, **n**. | 0.805 (.009) | 0.618 (.009) | 0.656 (.007) | 0.677 (.009) |
| CHI2, **n**. | 0.730 (.028) | 0.571 (.015) | 0.631 (.013) | 0.690 (.015) |
| ICP, **d**. | 0.816 (.006) | **0.622 (.004)** | 0.673 (.007) | **0.714 (.007)** |
| MR, **d**. | 0.766 (.011) | 0.602 (.006) | 0.651 (.009) | 0.711 (.008) |
| CHI2, **d**. | 0.815 (.010) | 0.597 (.007) | 0.671 (.007) | 0.705 (.006) |
| No transfer | 0.791 (.010) | 0.587 (.010) | 0.646 (.010) | 0.704 (.007) |
| MAML | 0.791 (.008) | 0.612 (.006) | 0.667 (.012) | 0.701 (.006) |
| TSA-MAML | 0.792 (.013) | 0.609 (.010) | 0.668 (.013) | 0.701 (.010) |
| DKT | 0.809 (.009) | 0.601 (.010) | 0.652 (.013) | 0.695 (.012) |
| HBM | 0.797 (.007) | 0.609 (.007) | 0.670 (.005) | 0.701 (.007) |

# 8. Conclusion

We have introduced a novel Bayesian meta-learning method that successfully leverages causal embeddings and expert-knowledge. Our approach uses the causal embeddings to set task-specific priors, while combining their information via a hierarchical Bayesian prior. Additionally, we provide theoretical guarantees over the generalization risk and the negative transfer, when including imperfect experts and causal discovery methods. The method we have proposed performs better under distribution shift than alternative methods, in synthetic cases and in a large multi-disease prediction task on the UK Biobank.

Natural extensions of our work could look into uncertainty propagation from the expert inference and the causal embeddings to the meta-learning model, as well as learning the causal embeddings jointly during the meta-training from the source data. Additionally, propagating the uncertainty of the expert and the causal discovery to the theoretical results would provide even more realistic guarantees over the generalization risk.

Although the motivation for our contributions stem from cross-disease transfer learning, the model is setting-agnostic and other applications could include, for instance, robotics. For example, tasks have some intrinsic similarity (e.g., "*open a door*", "*open a drawer*") and the expert can describe the similarity between different tasks for (say) a robot hand. In healthcare, our method is particularly suitable to rare and emerging diseases, where outcomes are too scarce for causal discovery but expert mechanistic knowledge is available. We note that a practical limitation of our approach is that it requires access to a domain expert whose notion of task similarity aligns with the causal embedding space, which is a reasonable assumption in clinical settings. Additionally, the quality of transfer depends on the causal discovery method used to construct the source task embeddings, as different methods capture different notions of task similarity. In practice, the choice of causal discovery method should be validated for the specific application domain.

Finally, theoretical extensions of our work could consider moving from the meta-learning framework to neural processes (Garnelo et al., 2018a;b; Nguyen & Grover, 2022); for instance, by incorporating the causal embeddings we have described into these processes. Additionally, amortization (e.g., Huang et al., 2026) of the user model could prove useful for deployment scenarios where the queries are actively incorporated into the model.

# Acknowledgements

This work was supported by EU funding ERC ODD-ML 101201120, the Research Council of Finland (Flagship programme: Finnish Center for Artificial Intelligence FCAI and decisions 359567 and 358958), EU Horizon 2020 (European Network of AI Excellence Centres ELISE, grant agreement 951847) and UKRI Turing AI World-Leading Researcher Fellowship (EP/W002973/1). We also acknowledge the computational resources provided by the Aalto Science-IT Project from Computer Science IT. This research has been conducted using the UK Biobank Resource under application number 77565.

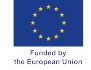 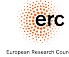

# Impact Statement

This paper presents a machine learning method that employs causal similarities based on data for transferring predictive disease models across disease tasks. Typical warnings about *proving* causality or *true* biological mechanisms through observational data apply to this method, which should be validated in a clinical setting or through randomized controlled trials. Moreover, we expect that our method will have a **positive** societal impact as it can improve generalization with sparse data, as such reducing data requirements for diseases with small cohorts (e.g., rare diseases). We further note that the method is robust to distribution shift, and we provide a mathematical formulation for negative transfer that we suspect is also a **positive** societal impact.

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

# A. Mathematical Derivations

In this section we provide the formal mathematical proofs which ensure that by moving in the causal space, through the embeddings ($z$), we are able to move the predictions to new tasks. This is important because it guarantees that the method is able to capture new behaviours by changing solely through the embedding space. Additionally, we provide a Lipschitz-continuity proof of the error. Lipschitz-continuity of the error implies that small changes in the causal space do not catastrophically affect the prediction.

## A.1. Proof of Proposition 4

We split the proof of Proposition 4 in two parts. First we demonstrate that the prior-induced risk is Lipschitz, and then we provide the error decomposition proof in Appendix A.1.2.

### A.1.1. THE PRIOR-INDUCED RISK IS LIPSCHITZ IN THE EMBEDDING SPACE

*Proof.* For each task $t$, let $p_t$ denote the data-generating distribution over $(x, y)$, which does not depend on the model parameters $\phi$. For predictor $\phi$, define the task risk

$$\mathcal{R}_t(\phi) = \mathbb{E}_{(x,y) \sim p_t}[\ell(\phi; x, y)], \qquad \bar{\mathcal{R}}_t(z) = \mathbb{E}_{\phi \sim p(\phi|z)}[\mathcal{R}_t(\phi)]$$

and consider the embedding-conditioned prior

$$p(\phi \mid z) = \mathcal{N}(\theta + Wz, \sigma^2 I).$$

Let $\ell(\phi; x, y) \in [0, M]$ be a bounded loss, and for $i = 1, 2$, denote by $p_i = p(\phi \mid z_i)$. Then, if $g(\phi) = M^{-1}\mathcal{R}_t(\phi)$, we have that

$$\begin{aligned}
\left|\bar{\mathcal{R}}_t(z_1) - \bar{\mathcal{R}}_t(z_2)\right| &= M \left|\mathbb{E}_{p_1}[g] - \mathbb{E}_{p_2}[g]\right| \\
&= M \left|\mathbb{E}_{p_1}[g] - \mathbb{E}_{p_2}[g]\right| \\
&\leq M D_{\mathrm{TV}}(p_1, p_2),
\end{aligned}$$

where the last line follows by definition of total variation distance (Section 2.4, pp. 83–84, Tsybakov, 2009).

Next, applying Pinsker's inequality (Section 2.4, pp. 88–89, Tsybakov, 2009), we obtain

$$D_{\mathrm{TV}}(p_1, p_2) \leq \sqrt{\tfrac{1}{2} D_{\mathrm{KL}}(p_1 \| p_2)}.$$

Since $p_1$ and $p_2$ share covariance $\sigma^2 I$, their KL divergence admits the closed form

$$D_{\mathrm{KL}}\big(p(\phi \mid z_1) \| p(\phi \mid z_2)\big) = \frac{1}{2\sigma^2} \|W(z_1 - z_2)\|^2 \leq \frac{\|W\|^2}{2\sigma^2} \|z_1 - z_2\|^2.$$

Applying the bound above yields:

$$\left|\bar{\mathcal{R}}_t(z_1) - \bar{\mathcal{R}}_t(z_2)\right| \leq \frac{M\|W\|}{2\sigma} \|z_1 - z_2\|.$$

Finally, since $\|z_1 - z_2\| \leq \varepsilon$, due to $\varepsilon$-similarity of $z_1, z_2$, we obtain the desired result:

$$\left|\bar{\mathcal{R}}_t(z_1) - \bar{\mathcal{R}}_t(z_2)\right| \leq \frac{M\|W\|}{2\sigma} \varepsilon.$$

$\square$

### A.1.2. ERROR DECOMPOSITION ON A TARGET TASK

*Proof.* Let $\hat{\phi}_{t'}^X$ denote the predictor obtained on a target task $t'$ by method $X$ and let $\mathcal{R}_{t'}(\hat{\phi}_{t'}^X)$ denote its target risk. For any embedding $z$, the prior-induced risk is given by

$$\bar{\mathcal{R}}_{t'}(z) = \mathbb{E}_{\phi \sim p(\phi|z)}[\mathcal{R}_{t'}(\phi)]$$

Let $\hat{z}_{t'}$ denote the estimated embedding used for target task $t'$ at meta-test time, and $\bar{z} = \frac{1}{|\mathcal{T}_{\text{source}}|} \sum_{t \in \mathcal{T}_{\text{source}}} z_t$ is the empirical mean of the source embeddings. The prior mismatch is

$$\mathcal{E}_{\text{prior}}(t') := |\bar{\mathcal{R}}_{t'}(\hat{z}_{t'}) - \bar{\mathcal{R}}_{t'}(\bar{z})|.$$

Denote by $z_{t'}$ the true latent causal embedding, $\tilde{z}_{t'}$ its causal discovery estimate, and $\hat{z}_{t'}$ the embedding used by the method, and define

$$\varepsilon_{\text{expert}} := \|\hat{z}_{t'} - \tilde{z}_{t'}\|, \quad \varepsilon_{\text{causal}} := \|\tilde{z}_{t'} - z_{t'}\|, \quad \varepsilon_{\text{OOD}} := \|z_{t'} - \bar{z}\|.$$

Using the result proved in Appendix A.1.1, we obtain

$$\mathcal{E}_{\text{prior}}(t') \leq \frac{M\|W\|}{2\sigma} \|\hat{z}_{t'} - \bar{z}\|.$$

Using the triangle inequality on $d_{\mathcal{Z}}$, with intermediate embeddings $z_{t'}, \tilde{z}_{t'}, \hat{z}_{t'}$, we get

$$\|\hat{z}_{t'} - \bar{z}\| \leq \underbrace{\|\hat{z}_{t'} - \tilde{z}_{t'}\|}_{\varepsilon_{\text{expert}}} + \underbrace{\|\tilde{z}_{t'} - z_{t'}\|}_{\varepsilon_{\text{causal}}} + \underbrace{\|z_{t'} - \bar{z}\|}_{\varepsilon_{\text{OOD}}}.$$

Combining the inequalities above the desired result is obtained:

$$\mathcal{E}_{\text{prior}}(t') \leq \frac{M\|W\|}{2\sigma}\Big(\varepsilon_{\text{expert}} + \varepsilon_{\text{causal}} + \varepsilon_{\text{OOD}}\Big).$$

$\square$

## A.2. Full Statement and Proof of Theorem 5

*Remark* 6. If the source embeddings are sampled from a distribution with mean 0, then $\bar{z} \to 0$, when the number of tasks $|\mathcal{T}_{\text{source}}| \to \infty$ by a standard application of the law of large numbers, making the $\varepsilon_{\text{OOD}} \approx \|z_{t'}\|$.

We start this section with a formal definition of negative transfer, following the definition of Wang et al. (2019). Next, we explicitly provide the two conditions that are stated as mild in the main text, and proceed with the proof of Theorem 5.

**Definition 7.** (Negative transfer, Wang et al., 2019) Let $\hat{\phi}_{t'}^{\text{NT}}$ denote a predictor trained on the target task $t'$ only (no transfer). For any transfer method $X$, the negative transfer is given by the difference in risks

$$\text{NT}_X(t') := \mathcal{R}_{t'}(\hat{\phi}_{t'}^X) - \mathcal{R}_{t'}(\hat{\phi}_{t'}^{\text{NT}}).$$

In order to prove Theorem 5, we require the following assumptions.

**Assumption 8.** The population risk minimizer for task $t'$ satisfies

$$\phi_{t'}^* := \arg\min_{\phi} \mathcal{R}_{t'}(\phi) = \theta + W z_{t'}.$$

Assumption 8 is meant to remind the reader of the low-rank adaptation assumption (LoRA; Hu et al., 2022), which for a new task there is a rank-one matrix that helps adapt to a new setting. We avoid the assumption of rank-one, and assume that $W$ is a full rank matrix, and as such it can adapt to multiple different tasks. We consider this to be a mild assumption; for instance, we are not assuming that $W$ can be easily decomposed, the standard in low-rank methods. Additionally, we note that standard meta-learning approaches assume that there is no such matrix for different tasks and instead (implicitly) assume that tasks which are causally related are i.i.d.

**Assumption 9.** There exists constants $0 < \kappa_0 < \kappa$ such that for method $X$

$$\kappa_0\|\phi_{t'}^{X,0} - \phi_{t'}^*\| \leq \|\hat{\phi}_{t'}^X - \phi_{t'}^*\| \leq \kappa\|\phi_{t'}^{X,0} - \phi_{t'}^*\|.$$

This assumption ensures that the initial starting point is neither arbitrarily close or arbitrarily far from the optimized value.

**Assumption 10** (Risk monotonicity)**.** The population risk $\mathcal{R}_{t'}(\phi)$ is increasing in $\|\phi - \phi_{t'}^*\|$ in a neighborhood of $\phi_{t'}^*$.

This assumption is a standard regularity condition ensuring that the risk will not behave too erratically around the optimal parameter. It is implied by local convexity of $\mathcal{R}_{t'}(\phi)$ at $\phi_{t'}^*$, but is strictly weaker.

**Theorem 5.** *Under Assumptions 8–10, stable adaptation, a well-conditioned embedding map $W$, and if*

$$\varepsilon_{\text{expert}} + \varepsilon_{\text{causal}} \leq C \cdot \varepsilon_{\text{OOD}},$$

*for a constant $C$. Then, the causal prior mitigates negative transfer relative to the global prior, i.e.,*

$$\text{NT}_{\text{causal}}(t') \leq \text{NT}_{\text{glob}}(t').$$

*Proof.* By definition of negative transfer,

$$\text{NT}_X(t') = \mathcal{R}_{t'}(\hat{\phi}_{t'}^X) - \mathcal{R}_{t'}(\hat{\phi}_{t'}^{\text{NT}}).$$

Comparing negative transfer under the causal and global priors with the predictors $\hat{\phi}_{t'}^{\text{causal}}$ and $\hat{\phi}_{t'}^{\text{glob}}$ yields

$$\begin{aligned}
\Delta_{\text{NT}}(t') &:= \text{NT}_{\text{causal}}(t') - \text{NT}_{\text{glob}}(t') \\
&= \left(\mathcal{R}_{t'}(\hat{\phi}_{t'}^{\text{causal}}) - \mathcal{R}_{t'}(\hat{\phi}_{t'}^{\text{NT}})\right) - \left(\mathcal{R}_{t'}(\hat{\phi}_{t'}^{\text{glob}}) - \mathcal{R}_{t'}(\hat{\phi}_{t'}^{\text{NT}})\right) \\
&= \mathcal{R}_{t'}(\hat{\phi}_{t'}^{\text{causal}}) - \mathcal{R}_{t'}(\hat{\phi}_{t'}^{\text{glob}}),
\end{aligned}$$

since the no-transfer baseline cancels.

Thus, showing that the causal prior mitigates negative transfer relative to the global prior, means we need to show that:

$$\mathcal{R}_{t'}(\hat{\phi}_{t'}^{\text{causal}}) \leq \mathcal{R}_{t'}(\hat{\phi}_{t'}^{\text{glob}}). \tag{12}$$

By Assumption 10, this reduces to showing

$$\|\hat{\phi}_{t'}^{\text{causal}} - \phi_{t'}^*\| \leq \|\hat{\phi}_{t'}^{\text{glob}} - \phi_{t'}^*\|.$$

Now, let $\phi_{t'}^{\text{causal},0} = \theta + W\hat{z}_{t'}$ and $\phi_{t'}^{\text{glob},0} = \theta$ denote the prior means. Under Assumption 8, the distances between the global prior and the causal prior to task optimum $\phi_{t'}^*$ are respectively

$$\|\phi_{t'}^{\text{glob},0} - \phi_{t'}^*\| = \|\theta - (\theta + Wz_{t'})\| = \|Wz_{t'}\|, \text{ and} \tag{13}$$

$$\|\phi_{t'}^{\text{causal},0} - \phi_{t'}^*\| = \|(\theta + W\hat{z}_{t'}) - (\theta + Wz_{t'})\| = \|W(\hat{z}_{t'} - z_{t'})\|. \tag{14}$$

Note that by Remark 6, the last equality of Equation (13) corresponds to $\varepsilon_{\text{OOD}}$. Additionally, the equality of Equation (14) is bounded by $\varepsilon_{\text{expert}} + \varepsilon_{\text{causal}}$, where the expert error is $\varepsilon_{\text{expert}} = \|\hat{z}_{t'} - \tilde{z}_{t'}\|$ and the causal discovery error is $\varepsilon_{\text{causal}} = \|\tilde{z}_{t'} - z_{t'}\|$.

Applying Assumption 9, we have the bounds

$$\kappa_0\|\phi_{t'}^{\text{glob},0} - \phi_{t'}^*\| \leq \|\hat{\phi}_{t'}^{\text{glob}} - \phi_{t'}^*\|, \qquad \|\hat{\phi}_{t'}^{\text{causal}} - \phi_{t'}^*\| \leq \kappa\|\phi_{t'}^{\text{causal},0} - \phi_{t'}^*\|.$$

This implies that $\|\hat{\phi}_{t'}^{\text{causal}} - \phi_{t'}^*\| \leq \|\hat{\phi}_{t'}^{\text{glob}} - \phi_{t'}^*\|$ whenever

$$\kappa\|\phi_{t'}^{\text{causal},0} - \phi_{t'}^*\| \leq \kappa_0\|\phi_{t'}^{\text{glob},0} - \phi_{t'}^*\|$$

Substituting Equations (13) and (14) we get

$$\kappa\|W(\hat{z}_{t'} - z_{t'})\| \leq \kappa_0\|Wz_{t'}\|.$$

Since $W$ is well-conditioned, this is simplified to

$$\|\hat{z}_{t'} - z_{t'}\| \leq \frac{\kappa_0}{\kappa}\|z_{t'}\|,$$

a sufficient condition for the causal prior to be closer to $\phi_{t'}^*$ than the global prior.

Defining $\varepsilon_{\text{OOD}} := \|z_{t'}\|$ using Remark 6, this yields

$$\varepsilon_{\text{expert}} + \varepsilon_{\text{causal}} \leq C \cdot \varepsilon_{\text{OOD}},$$

with constant $C = \frac{\kappa_0}{\kappa}$. The desired result, Equation (12), follows. □

# B. Full Algorithms

## B.1. Meta-Learning

We describe the full meta-training and meta-testing algorithms used in our experiments. Our approach builds on hierarchical Bayesian meta-learning formulations, namely Ravi & Beatson (2019), but is implemented independently and extended to incorporate task-dependent priors via causal task embeddings. The losses considered are:

$$\mathcal{L}_1(\psi_t, \mathcal{D}_t^{(s)}, z_t) := -\mathbb{E}_{q_{\psi_t}(\phi_t \mid \mathcal{D}_t^{(s)})}\Big[\log p(\mathbf{y}_t^{(s)} \mid \mathbf{x}_t^{(s)}, \phi_t)\Big] + D_{\mathrm{KL}}\Big(q_{\psi_t}(\phi_t \mid \mathcal{D}_t^{(s)}) \,\|\, p(\phi_t \mid z_t, \theta)\Big), \qquad (15)$$

$$\mathcal{L}_2(\psi_t, \mathcal{D}_t^{(q)}, \lambda) := -\mathbb{E}_{q_{\psi_t}(\phi_t \mid \mathcal{D}_t^{(s)})}\Big[\log p(\mathbf{y}_t^{(q)} \mid \mathbf{x}_t^{(q)}, \phi_t)\Big] + D_{\mathrm{KL}}\Big(q_\lambda(\theta) \,\|\, p(\theta)\Big). \qquad (16)$$

During meta-training (Algorithm B.1), we sample a mini-batch of source tasks and split data of each task $\mathcal{D}_t$ into support $\mathcal{D}_t^{(s)}$ and query sets $\mathcal{D}_t^{(q)}$. We perform $K$ inner-loop stochastic variational updates on $\mathcal{D}_t^{(s)}$ by minimizing $\mathcal{L}_1$ (Equation (15)) starting from embedding conditional prior to obtain a variational task posterior $\psi_t^{(k+1)}$ (Line B.1). The outer loop then updates both the embedding weights $W$ and the global variational posterior $\lambda$ (Lines B.1 and B.1), by minimizing $\mathcal{L}_2$ (Equation (16)) across tasks, including $\ell_2$ regularization on $W$.

---

**Algorithm B.1** Causally-Aware Meta-Learning: meta-training

---

**Require:** Task distribution $p(\mathcal{T})$; learning rates $\beta_\theta, \beta_\phi$; inner steps $K$; mini-batch sizes $M_{\text{tasks}}, M_{\text{samples}}$
**Require:** Task embeddings $\{z_t\}_{t \in \mathcal{T}}$, embedding prior weights $W$, regularization coefficient $\gamma_W$
1: **Initialize:** global posterior parameters $\lambda = (\mu_\lambda, \sigma_\lambda^2)$, embedding weights $W$
2: **while** not done **do**
3:     Sample mini-batch of $M_{\text{tasks}}$ tasks $t \sim p(\mathcal{T})$
4:     **for all** tasks $t$ in mini-batch **do**
5:         Sample $M_{\text{samples}}$ from $\mathcal{D}_t$ and split into support $\mathcal{D}_t^{(s)}$ and query $\mathcal{D}_t^{(q)}$
6:         Compute: $\mu_t = \mu_\lambda + W z_t$
7:         Initialize task posterior $\psi_t^{(0)}$ from embedding-conditional prior $q_{(\mu_t, \sigma_t^2)}(\theta)$
8:         **for** $k = 0$ to $K - 1$ **do**
9:             $\psi_t^{(k+1)} \leftarrow \psi_t^{(k)} - \beta_\phi \nabla_{\psi_t^{(k)}} \mathcal{L}_1(\psi_t^{(k)}, \mathcal{D}_t^{(s)}, \mu_t, \sigma_t^2)$ {Inner loop adaptation}
10:         **end for**
11:     **end for**
12:     Update global posterior and embedding parameters {Outer-loop adaptation}
13:     $W \leftarrow W - \beta_W \nabla_W \frac{1}{M_{\text{tasks}}} \sum_t \mathcal{L}_2(\psi_t^{(K)}, \mathcal{D}_t^{(q)}, \mu_t, \sigma_t^2) + \gamma_W \|W\| t_2^2$
14:     $\lambda \leftarrow \lambda - \beta_\theta \nabla_\lambda \frac{1}{M_{\text{tasks}}} \sum_t \mathcal{L}_2(\psi_t^{(K)}, \mathcal{D}_t^{(q)}, \mu_t, \sigma_t^2)$
15: **end while**

---

For the meta-test approach (Algorithm B.2), we split the target task data $\mathcal{D}_{t'}$ into adaptation $\mathcal{D}_{t'}^{\text{tr}}$ and test sets $\mathcal{D}_{t'}^{\text{ts}}$. We initialize the prior for target task $t'$ from the global training posterior $\lambda$, conditioned on the target task embedding $z_{t'}$, and adapt in the inner loop (Line B.2) on target adaptation data $\mathcal{D}_{t'}^{\text{tr}}$ by minimizing $\mathcal{L}_1$ (Equation (15)), making the final predictions on the held-out target data.

---

**Algorithm B.2** Causally-Aware Meta-Learning: meta-testing

---

**Require:** Target tasks $\mathcal{T}_{\text{target}}$, task embeddings $z_{t'}$, global training posterior $\lambda$, learned embedding weights $W_\mu$, inner steps $K$, learning rate $\beta_\phi$

1: **for all** target tasks $t' \in \mathcal{T}_{\text{target}}$ **do**
2:      Split $\mathcal{D}_{t'}$ into adaptation set $\mathcal{D}_{t'}^{\text{tr}}$ and test set $\mathcal{D}_{t'}^{\text{ts}}$
3:      Compute: $\mu_{t'} = \mu_\lambda + W_\mu z_{t'}$
4:      Initialize task posterior $\psi_{t'}^{(0)} = (\mu_{t'}, \sigma_{t'}^2)$
5:      **for** $k = 0$ to $K - 1$ **do**
6:          Sample mini-batch of $M_{\text{samples}}$ from $\mathcal{D}_{t'}^{\text{tr}}$
7:          $\psi_{t'}^{(k+1)} \leftarrow \psi_{t'}^{(k)} - \beta_\phi \nabla_{\psi_{t'}^{(k)}} \mathcal{L}_1(\psi_{t'}^{(k)}, \mathcal{D}_{t'}^{\text{tr}}, \mu_{t'}, \sigma_{t'}^2)$   {Inner loop adaptation}
8:      **end for**
9:      Use final $\psi_{t'}^{(K)}$ for predictions on $\mathcal{D}_{t'}^{\text{ts}}$
10: **end for**

---

## B.2. Expert Feedback Model

In this section, we describe the expert feedback model (Algorithm B.3) introduced in Section 5. For each query $b = 1, \ldots, B$, we select a most informative pair of source tasks using Bayesian active learning by disagreement (BALD), where entropies $H[\cdot]$ are approximated via Monte Carlo samples from the current variational posterior $q_\varphi(z_{t'})$. We query the expert to determine which source task is closer to the target and add the observed response to the comparison set $\mathcal{C}$ before updating a variational posterior over the target task embedding $z_{t'}$ via stochastic variational inference. After $B$ queries, the posterior mean of the inferred embedding is returned.

---

**Algorithm B.3** Expert-guided inference of target task embedding

---

**Require:** Source task embeddings $\{z_i\}_{i \in \mathcal{T}_{\text{source}}}$, target task $t'$, query budget $B$

1: Initialize prior $p(z_{t'}) = \mathcal{N}(0, I_d)$
2: Initialize comparison set $\mathcal{C} \leftarrow \emptyset$
3: **for** $b = 1, \ldots, B$ **do**
4:      Select query $\xi_b = (i_b, j_b)$ with BALD

$$\xi_b = \underset{(i,j)}{\arg\max}\, H[y_{ij} \mid \mathcal{C}] - \mathbb{E}_{z_{t'} \sim q_\varphi}[H[y_{ij} \mid z_{t'}]].$$

5:      Query expert: is $i_b$ or $j_b$ closer to $t'$?
6:      Receive response $c_b \in \{0, 1\}$.
7:      $\mathcal{C} \leftarrow \mathcal{C} \cup \{(\xi_b, c_b)\}$.
8:      Update $q_\varphi(z_{t'})$ via SVI with likelihood $p(c_b \mid z_{t'}, \xi_b) = \Phi(\tau \cdot \Delta(\xi_b; z_{t'}))$.
9: **end for**
10: **Return** $\hat{z}_{t'} = \mathbb{E}_q[z_{t'} \mid \mathcal{C}]$.

---

# C. Additional Experiments

In this section we present additional experimental results.

## C.1. Imperfect Causal Discovery

We study the effect of imperfect causal discovery on meta-learning performance, by constructing noisy versions of the causal task embeddings. For each task task embedding $z_t$, we sample a task-specific perturbation direction $d_t \sim \mathcal{N}(0, I_4)$ and set $u_t = d_t/\|d_t\|$. With a noise level $\sigma_c = \{0, 0.5, 0.8\}$, we define

$$\tilde{z}_t = \|z_t\| \cdot \frac{z_t + \sigma_c \|z_t\| u_t}{\|z_t + \sigma_c \|z_t\| u_t\|}.$$

This construction introduces directional misalignment in the embedding space while preserving the embedding scale. The resulting embeddings are used to condition the task-specific priors in the meta-learning while the data-generating process remains unchanged. The results are presented in Figure 4, where AUROC as a function of $\varepsilon_{\mathrm{OOD}}$ is presented. We see that with moderate noise $\sigma_c = 0.5$ the performance of meta-learning stays robust under task-shift and starts to degrade only under high noise $\sigma_c = 0.8$.

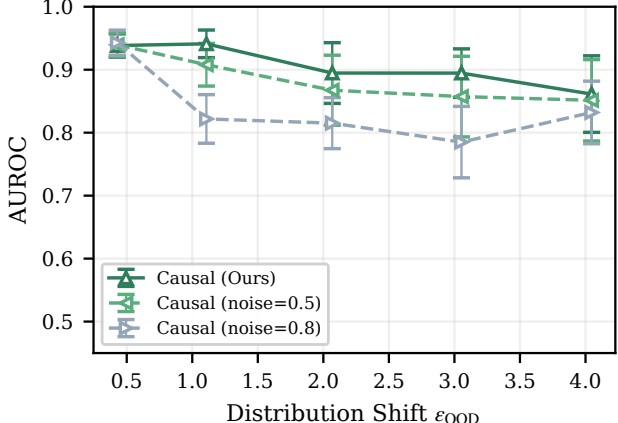

*Figure 4.* Performance of causally-aware meta-learning models under increasing noise in the causal task embeddings.

## C.2. Partial Causal Graph

We study the effect of incomplete causal discovery, where only a subset of the true causal parents are identified. We simulate a setting where $k$ out of $d_z = 4$ causal parents are identified and other dimensions are set to zero. The results are presented in Figure 5, where we compare no causal information (zero embeddings), partial causal information ($k = 2$, i.e., $50\%$ of parents identified), and full causal information. We observe that even with only half of the parents identified, the method outperforms the no-causal baseline across all OOD levels.

## C.3. Nonlinear Prior Ablation

We compare the linear parametrization against a nonlinear variant $\phi_t = \theta + \sigma(W z_t)$ on the synthetic data, where $\sigma$ is a MLP consisting of a linear layer with hidden dimension 16, followed by ReLU activation and a linear output layer. Figure 6 shows that the linear parametrization outperforms the nonlinear variant, and the nonlinear variant shows intermediate performance between the causal linear and the global prior (HBM).

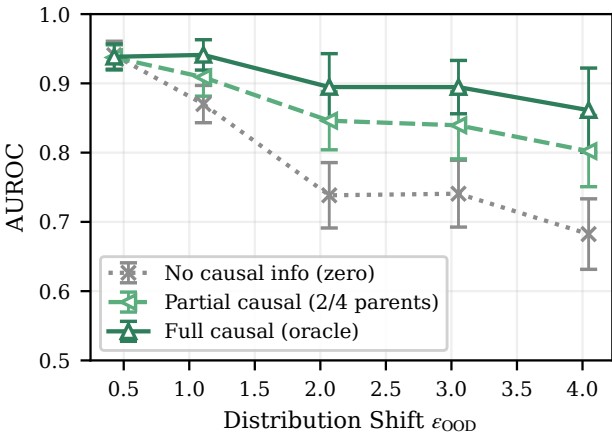

*Figure 5.* Performance of causally-aware meta-learning with partial causal information.

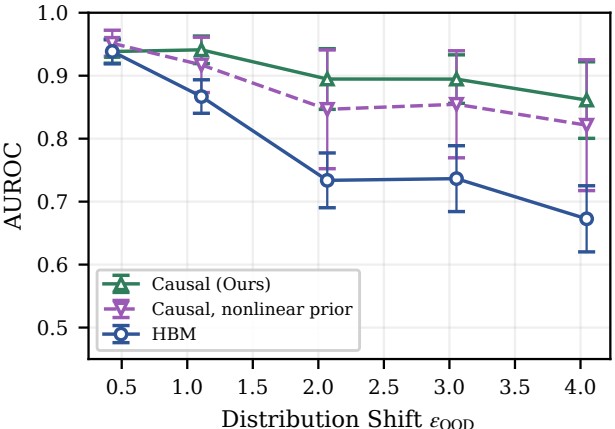

*Figure 6.* Comparison of linear vs. nonlinear prior parametrization on synthetic data.

## C.4. Expert Model Sensitivity to Expert Noise

In this experiment, we assess the sensitivity of the expert model to expert unreliability. To simulate an unreliable expert, we vary the $\tau_{\text{expert}}$ parameter in the likelihood (Eq. (8)) that is used to simulate the expert answers. Figure 7 shows the RMSE between inferred and true task embeddings as a function of the number of expert queries for target tasks ordered by increasing distribution shift for $\tau_{\text{expert}} = \{0.5, 1.0, 2.0\}$, with higher values corresponding to a more reliable expert. Across all tasks, lower expert noise leads to faster convergence and lower final error. For task 20 (magnitude of distribution shift $s = 0.01$), the true task embedding lies close to the prior mean, explaining the almost zero RMSE before expert querying.

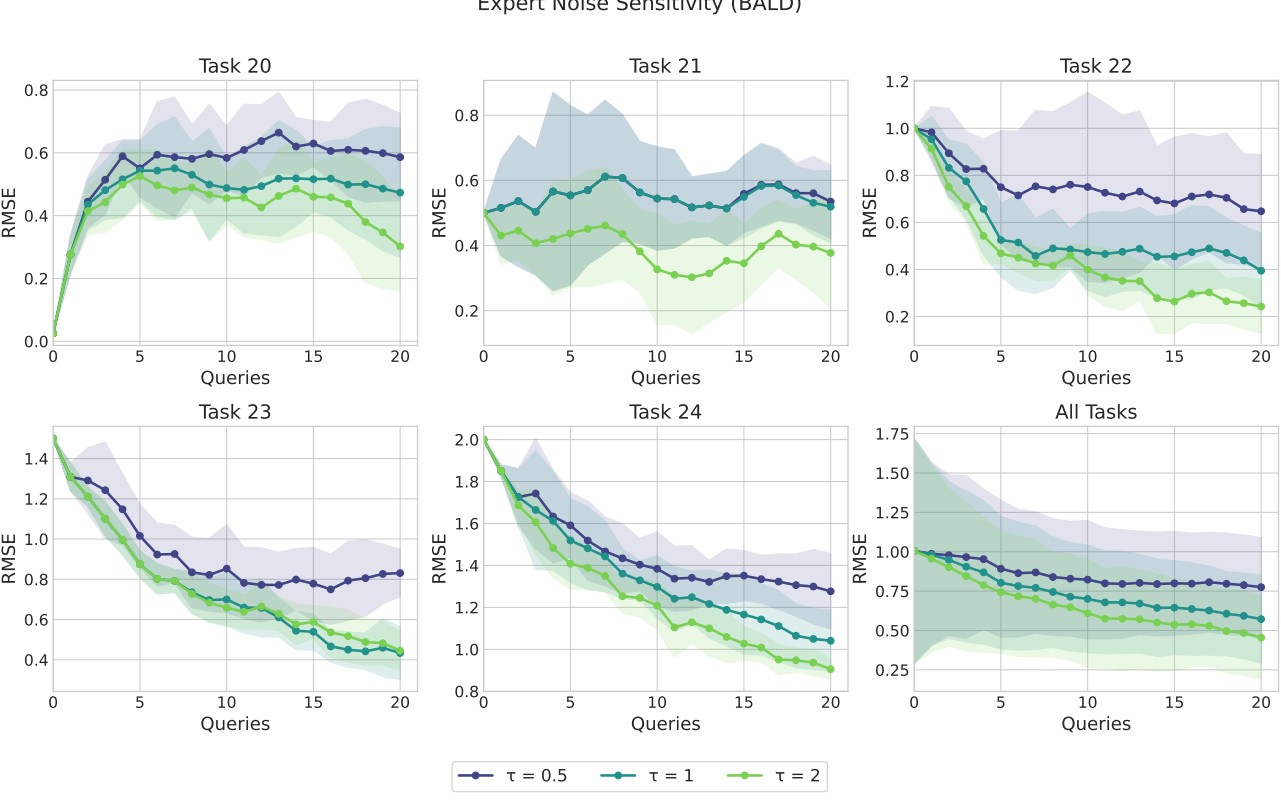

*Figure 7.* Expert-noise sensitivity under active querying. RMSE as a function of the number of expert queries at different noise levels $\tau_{\text{expert}}$ for multiple target tasks ordered with increasing distribution shift. Lower $\tau_{\text{expert}}$ corresponds to noisier expert feedback. The final panel aggregates results across all target tasks.

## C.5. Active Query Strategy vs Random

In this experiment, we compare active query strategy with BALD against random querying, to assess if active querying leads to more accurate inference with smaller query budget. Figure 8 shows the RMSE between inferred and true task embeddings as a function of number of expert queries for target tasks ordered by increasing distribution shift for both query strategies at $\tau_{\text{expert}} = 1.0$. We see that active querying achieves faster reductions in embedding error and lower final RMSE across tasks at high level of task shift. At low shift (tasks 20 and 21), the performance is similar active vs random, indicating that in in-distribution regimes, all queries are similarly informative.

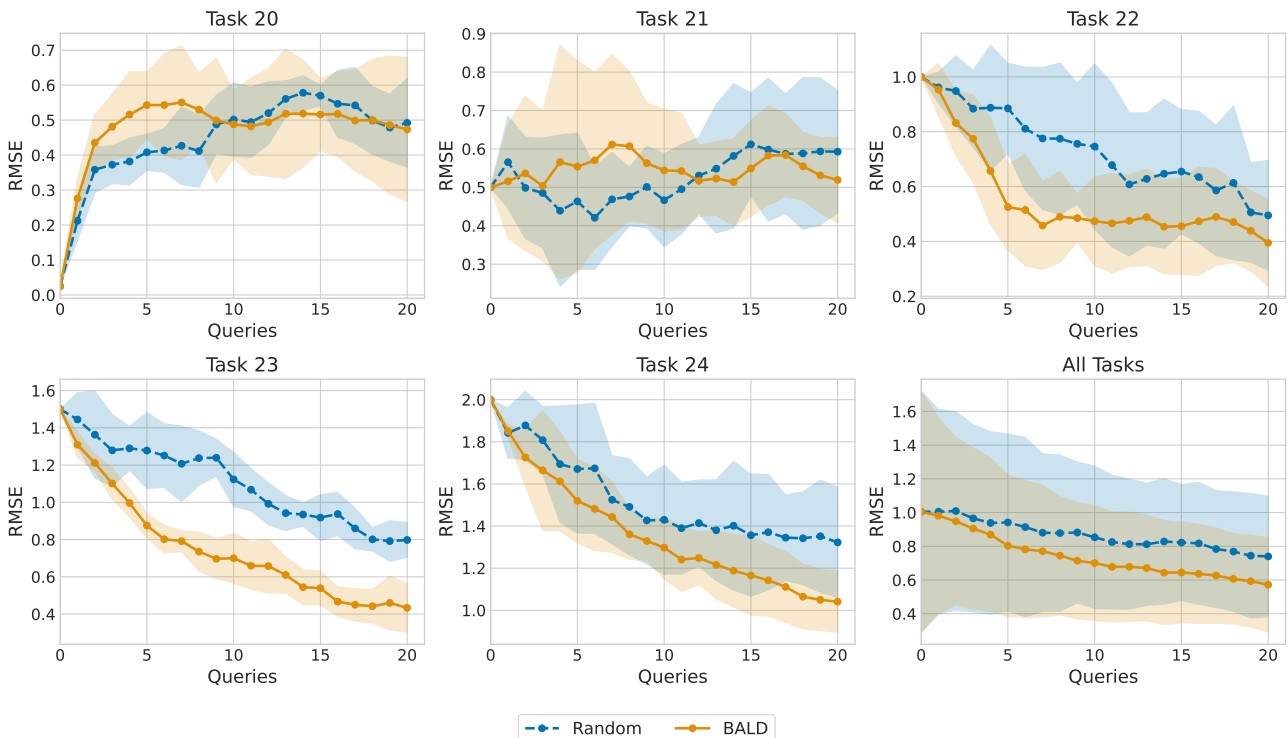

*Figure 8.* Random vs. active (BALD) querying. RMSE as a function of the number of expert queries for multiple target tasks ordered with increasing distribution shift, $\tau_{\text{expert}} = 1.0$. The final panel aggregates results across all target tasks.

# D. Experimental Details

In this section we describe the implementation details of our method and the other baselines, as well as the experimental settings for our synthetic experiment and the application to the cross-disease transfer in the UK Biobank.

## D.1. Data Generation for Synthetic Experiments

Following the notation from Section 2.2, for each task $t \in \mathcal{T} = \mathcal{T}_{\text{source}} \cup \mathcal{T}_{\text{target}}$ we simulate a dataset $\mathcal{D}_t = \{(\mathbf{x}_{t,i}, y_{t,i})\}_{i=1}^{500}$, by using a (task-specific) structural causal model given by $\mathcal{W}_t = (\mathbf{V}, \mathbf{U}, F_t, P_U)$. All tasks share the endogenous variables $\mathbf{V} = \{X_1, \ldots, X_{10}\}$ and the causal structure, but differ in their structural equations $F_t$. The values of $X_i$ are generated by sampling $\mathbf{x}_{t,i} \overset{i.i.d.}{\sim} \mathcal{N}(0, I_{10})$. For each task we assign a latent causal task embedding $z_t \in \mathcal{Z} \subseteq \mathbb{R}^4$, that parameterizes the task-specific data-generating mechanism. Source tasks ($|\mathcal{T}_{\text{source}}| = 20$) are sampled from a centered distribution $z_t \overset{i.i.d.}{\sim} \mathcal{N}(0, (0.8)^2 I_4)$, for $t \in \mathcal{T}_{\text{source}}$. Target tasks $t'$ ($|\mathcal{T}_{\text{target}}| = 5$) are generated by moving along a fixed unit vector $\delta \in \mathbb{R}^4$ with $\|\delta\| = 1$, $z_{t'} = s \cdot \delta$, for $t' \in \mathcal{T}_{\text{target}}$, where $s \in \{0.1, 1.0, 2.0, 3.0, 4.0\}$ controls the magnitude of distribution shift, with smaller values corresponding to smaller shifts.

To generate binary outcome variables $Y_t$, we first generate the parent set $\mathcal{P} = \{1, 2, 3, 4\}$ (i.e., features $X_1, \ldots, X_4$), and intermediate continuous $Y_t^{(c)}$, by

$$Y_t^{(c)} = \sum_{j \in \mathcal{P}} X_j \, e_{t,j} + U_Y; \ U_Y \overset{i.i.d.}{\sim} \mathcal{N}(0, (0.6)^2).$$

The task-specific causal effects $e_t \in \mathbb{R}^{10}$ are generated by

$$e_t = b + W z_t + \eta_t, \qquad \eta_t \overset{i.i.d.}{\sim} \mathcal{N}(0, (0.15)^2 I).$$

The shared weight matrix $W \in \mathbb{R}^{10 \times 4}$ has rows $W_j \overset{i.i.d.}{\sim} \mathcal{N}(0, (0.5)^2 I_4)$, for parent features $j \in \mathcal{P}$ and $W_j = 0$ otherwise. The bias vector $b \in \mathbb{R}^{10}$ has entries $b_j \overset{i.i.d.}{\sim} \text{Unif}(0.5, 1.0)$, for $j \in \mathcal{P}$, and $b_j = 0$ otherwise. For non-parent features, $j \notin \mathcal{P}$, we set $e_{t,j} = 0$.

To model realistic dataset bias and shortcut learning, we include the spurious feature $X_{10}$ with $10 \notin \mathcal{P}$:

$$X_{10} = \alpha(s) \cdot (2Y_t - 1) + \epsilon, \qquad \epsilon \overset{i.i.d.}{\sim} \mathcal{N}(0, 1),$$

where $\alpha(s) = 0.5 \cdot (1 - s/s_{\max})$ with $s_{\max} = 4.0$ for target tasks, and $\alpha = 0.4$ for source tasks (resulting in a correlation of approximately 0.3). $X_{10}$ is an observed feature present in every task, but it is not a causal parent of $Y_t$. Thus, it does not enter the structural equation for $Y_t^{(c)}$, and the spurious correlation decays with distribution shift magnitude $s$ while the causal mechanisms remains unchanged.

The final outcomes are obtained by thresholding at the 70th percentile of the continuous distribution of $Y_t^{(c)}$ (per task). The values above the threshold give $Y_t = 1$ and below give $Y_t = 0$, resulting in approximately 30% of positive labels per task.

## D.2. Expert Inference Model Implementation Details

In this section, we explain the implementation details of our expert inference model. A standard Gaussian prior $p(z_{t'}) = \mathcal{N}(0, I)$ is placed over the target task embedding, and we fix temperature $\tau = 1.0$. For experiments, simulated expert responses may be generated with a separate noise parameter $\tau_{\text{expert}}$ to assess robustness to model mismatch. We use query budget $B = 20$ unless otherwise stated. The expert inference model is implemented in Pyro (Bingham et al., 2018) and uses stochastic variational inference with diagonal Gaussian variational family (AutoDiagonalNormal) and the Trace ELBO objective with Adam optimizer. We use a learning rate of 0.01, perform 150 SVI steps after each expert query, and use 200 Monte Carlo samples to approximate the BALD acquisition function. All experiments use fixed random seeds.

## D.3. Meta-Learning and Baseline Implementation Details

In this section we detail the implementation details of our method and baseline models. The complete implementation can be found in Supplementary material.

**Shared neural network architecture.** The underlying predictive model for all methods is a Bayesian neural network (BNN) based on the long short-term memory (LSTM; Hochreiter & Schmidhuber, 1997) architecture. The architecture consists of two parallel networks with an LSTM layer for longitudinal data, and a fully connected multilayer perceptron (MLP) for tabular data. The outputs of the two networks are concatenated and passed through a linear layer, which produces logits to use in binary classification. Unless otherwise stated, all models use two hidden layers, with layer size of 32 for both MLP and LSTM components.

**Hierarchical meta-learning models.** Both meta-learning models are implemented in PyTorch. Task-specific and global model parameters are updated using diagonal variational inference operators implemented with the TorchOpt (Ren et al., 2023) and Posteriors (Duffield et al., 2025) libraries. The parameters of the embedding weight matrix $W$ are optimized separately using the Adam optimizer.

To mitigate known optimization instabilities in gradient-based meta-learning, we apply several standard stabilization strategies. The embedding-conditioned prior adaptations are norm-constrained relative to the corresponding global parameter norms via an adaptation scale $\alpha$, and only the prior mean is adapted while the variance is kept fixed. Embedding conditioned weights in $W$ are initialized to zero and regularized using $\ell_2$ penalty with regularization coefficient $\gamma_W$, with additional gradient clipping ($\|\nabla_W\|_2 \leq 1.0$) and weight-norm constraints ($\|W\|_2 \leq 0.5$). During the variational inference, KL-divergence terms are scaled to account for mini-batching, and we employ tempered posteriors with temperature hyperparameters $T_1, T_2$.

**Bayesian neural network baseline.** For a no-transfer baseline, we train a separate BNN independently for each target task using the same predictive architecture described above. Each model is trained from scratch using variational inference, without any parameter sharing across tasks. Similar to the hierarchical meta-learning models, the BNN baseline is optimized using diagonal variational inference with TorchOpt and Posteriors libraries.

**Model agnostic meta-learning baseline.** We are using gradient-based Model Agnostic Meta-Learning (MAML; Finn et al., 2017), following the implementation in the original paper modified to work with our data loader and shared neural network architecture.

**Deep kernel transfer learning baseline.** We are using deep kernel transfer learning (DKT; Patacchiola et al., 2020), following the implementation in the original paper modified to work with our data loader and shared neural network architecture.

### D.4. Synthetic Experimental Settings

The hyperparameters used for our synthetic experiments are detailed in Table 2. For all methods, task data are split into training (70% of samples) and test sets (30% of samples). The remaining data is further split into support (adaptation) and query (validation) datasets with 50-50% split. We use two-fold cross validation for all methods. For MAML and DKT we use 100 samples per task. Bayesian predictive quantities are approximated using Monte Carlo sampling with $S = 10$ posterior samples. The convergence of the models is verified using early stopping based on validation AUROC. All reported results are averaged over multiple runs with fixed random seeds.

*Table 2.* Final hyperparameter values used in synthetic experiments.

| Method | Hyperparameter | Value |
|---|---|---|
| BNN | Inner learning rate | $3 \times 10^{-3}$ |
| | Inner temperature | $10^{-1}$ |
| | Global prior std. $\sigma$ | 1.0 |
| | Model init. log std. | $-1$ |
| MAML | Inner learning rate | $3 \times 10^{-2}$ |
| | Outer learning rate | $3 \times 10^{-3}$ |
| | Num. inner steps | 4 |
| | Tasks per meta-batch | 4 |
| | Samples per iteration | 100 |
| TSA-MAML | Inner learning rate | $3 \times 10^{-2}$ |
| | Outer learning rate | $3 \times 10^{-3}$ |
| | Num. inner steps | 4 |
| | Tasks per meta-batch | 4 |
| | Samples per iteration | 100 |
| | Num. clusters | 3 |
| DKT | Outer learning rate | $10^{-3}$ |
| | GP optimization steps | 50 |
| | Tasks per meta-batch | 4 |
| | Samples per iteration | 100 |
| HBM | Inner learning rate | $10^{-4}$ |
| | Inner temperature | $5 \times 10^{-4}$ |
| | Global prior std. $\sigma$ | 0.05 |
| | Prior scaling | $10^{4}$ |
| | Model init. log std. | $-3$ |
| | Num. inner updates | 4 |
| | Outer learning rate | $10^{-3}$ |
| | Outer temperature | $5 \times 10^{-4}$ |
| Causal Meta-Learning | Inner learning rate | $10^{-4}$ |
| | Inner temperature | $5 \times 10^{-4}$ |
| | Global prior std. $\sigma$ | 0.05 |
| | Prior scaling | $10^{4}$ |
| | Model init. log std. | $-3$ |
| | Num. inner updates | 4 |
| | Outer learning rate | $10^{-3}$ |
| | Outer temperature | $5 \times 10^{-4}$ |
| | $W$ learning rate | $3 \times 10^{-4}$ |
| | $W$ regularizer $\gamma_W$ | $10^{-1}$ |
| | Adaptation scale $\alpha$ | 0.12 |

## D.5. UKBB Dataset and Experimental Settings

For the experiments in Section 7 we use the UK Biobank (with project permission 77565). Next, we detail the data processing to construct the specific dataset used.

**Dataset construction.** The UK Biobank data includes 220,571 individuals with comprehensive primary care and hospital records. Baseline covariates comprised age, sex, ordinal lifestyle indicators (BMI category, smoking status, alcohol consumption), and continuous blood biochemistry measurements. Lifestyle indicators and continuous blood biochemistry variables were obtained at baseline, corresponding to the year of participant recruitment into UK Biobank. Individuals with missing lifestyle responses were excluded from further analyses, and missing biochemistry values were imputed using the cohort median. Summary statistics of the dataset are presented in Table 3.

In addition, we constructed a longitudinal data set for the same patient cohort comprising of 304 diagnosis and medication variables, including ICD-10 diagnosis codes and medication codes encoded using the British National Formulary (BNF) and Read coding systems. These variables were aggregated at yearly intervals over the years 1990-2017. Each task was formulated as a binary classification problem, aiming to predict the occurrence of future outcomes after a fixed index year (2011) based on covariates observed prior to the index year.

*Table 3.* Baseline characteristics of the UK Biobank cohort. Panel A reports demographic and lifestyle characteristics, and Panel B reports blood biochemistry measurements, along with their respective units of measurement. Continuous variables are reported as mean $\pm$ SD. Categorical variables are reported in counts (percentages).

**A: Demographic and lifestyle characteristics**

| | |
|---|---|
| Participants, $N$ | 220,571 |
| Age (years) | 59.4 $\pm$ 8.1 |
| **Sex** | |
| Female | 121,370 (55.0%) |
| Male | 99,201 (45.0%) |
| **BMI category** | |
| Underweight | 1,059 (0.5%) |
| Normal weight | 67,814 (30.7%) |
| Overweight | 94,862 (43.0%) |
| Obese | 56,836 (25.8%) |
| **Smoking status** | |
| Never | 121,399 (55.0%) |
| Previous | 76,126 (34.5%) |
| Current | 23,046 (10.4%) |
| **Alcohol consumption** | |
| Never | 17,856 (8.1%) |
| Low / occasional | 50,146 (22.7%) |
| Moderate / high | 152,569 (69.2%) |

**B: Blood biochemistry measurements**

| | | |
|---|---|---|
| Alanine aminotransferase | (U/L) | 23.4 $\pm$ 13.8 |
| Alkaline phosphatase | (U/L) | 83.5 $\pm$ 25.1 |
| Apolipoprotein A | (g/L) | 1.53 $\pm$ 0.25 |
| Apolipoprotein B | (g/L) | 1.03 $\pm$ 0.23 |
| Aspartate aminotransferase | (U/L) | 26.1 $\pm$ 10.2 |
| C-reactive protein | (mg/L) | 2.52 $\pm$ 4.21 |
| Calcium | (mmol/L) | 2.38 $\pm$ 0.09 |
| Cholesterol | (mmol/L) | 5.70 $\pm$ 1.12 |
| Creatinine | ($\mu$mol/L) | 72.1 $\pm$ 17.9 |
| Cystatin C | (mg/L) | 0.91 $\pm$ 0.17 |
| Gamma glutamyltransferase | (U/L) | 36.7 $\pm$ 39.8 |
| Glucose | (mmol/L) | 5.10 $\pm$ 1.15 |
| HbA1c | (mmol/mol) | 36.1 $\pm$ 6.6 |
| HDL cholesterol | (mmol/L) | 1.44 $\pm$ 0.35 |
| IGF-1 | (nmol/L) | 21.4 $\pm$ 5.5 |
| LDL cholesterol | (mmol/L) | 3.56 $\pm$ 0.85 |
| Phosphate | (mmol/L) | 1.16 $\pm$ 0.15 |
| SHBG | (nmol/L) | 50.5 $\pm$ 25.6 |
| Total protein | (g/L) | 72.5 $\pm$ 3.8 |
| Triglycerides | (mmol/L) | 1.74 $\pm$ 1.00 |
| Urate | ($\mu$mol/L) | 308.2 $\pm$ 78.0 |
| Urea | (mmol/L) | 5.40 $\pm$ 1.35 |
| Vitamin D | (nmol/L) | 48.0 $\pm$ 20.0 |

**Task selection.** We select a group of diseases comprising of cardiovascular diseases and respiratory conditions, which are common in the UK Biobank and exhibit heterogeneous, but related, disease mechanisms. Tasks are split into source and target sets, where source tasks are used for meta-training and target tasks are held out for adaptation, as explained in Appendix B. Key details of the selected source and target tasks are summarized in Table 4.

*Table 4.* Overview of disease tasks used in the experiments, including ICD-10 codes, disease category (cardiovascular-metabolic = C or respiratory = R), clinical descriptions, task roles, and prevalence in the UK Biobank cohort (N=220,571).

| ICD-10 | Category | Disease description | Role | Cases | Prevalence (%) |
|---|---|---|---|---|---|
| I21 | C | Acute myocardial infarction | Target | 3,941 | 1.79 |
| G45 | C | Transient ischaemic attack (TIA) | Target | 2,399 | 1.09 |
| J44 | R | Chronic obstructive pulmonary disease | Target | 5,850 | 2.65 |
| J45 | R | Asthma | Target | 4,318 | 1.96 |
| E11 | C | Type 2 diabetes mellitus | Source | 9,571 | 4.34 |
| I10 | C | Essential (primary) hypertension | Source | 32,153 | 14.58 |
| I20 | C | Angina pectoris | Source | 5,325 | 2.41 |
| I25 | C | Chronic ischaemic heart disease | Source | 11,175 | 5.07 |
| I48 | C | Atrial fibrillation and flutter | Source | 9,704 | 4.40 |
| I63 | C | Cerebral infarction (ischaemic stroke) | Source | 2,898 | 1.31 |
| J18 | R | Pneumonia, organism unspecified | Source | 8,385 | 3.80 |
| J40 | R | Bronchitis, not specified acute/chronic | Source | 1,023 | 0.46 |
| J43 | R | Emphysema | Source | 1,474 | 0.67 |

**Experimental settings.** Bayesian optimization of the hyperparameters was done with Weights&Biases, optimizing mean validation AUROC across target tasks. To ensure fair comparison, all methods were given comparable tuning budgets and identical data splits. Model architectures were fixed across methods (number of layers=2, number of hidden layers=32 in both tabular and longitudinal sets). All methods were tuned over 50 sweeps. The final experiments were done with the best hyperparameter configuration evaluated across 10 random seeds. The used ranges for hyperparameters are presented in Table 5.

*Table 5.* Hyperparameter tuning ranges.

| Method | Hyperparameter | Search Range |
|---|---|---|
| BNN | Inner learning rate | log-uniform $[3 \times 10^{-5}, 3 \times 10^{-3}]$ |
| | Inner temperature | log-uniform $[10^{-4}, 3 \times 10^{1}]$ |
| | Global prior std. $\sigma$ | log-uniform $[0.02, 0.2]$ |
| | Prior scaling | log-uniform $[10^{2}, 10^{4}]$ |
| | Model init. log std. | $\{-4, -3, -2\}$ |
| MAML | Inner learning rate | log-uniform $[10^{-4}, 5 \times 10^{-2}]$ |
| | Outer learning rate | log-uniform $[10^{-5}, 5 \times 10^{-3}]$ |
| | Num. inner steps | $\{1, 2, 4, 8\}$ |
| | Tasks per meta-batch | $\{2, 4, 8\}$ |
| TSA-MAML | Inner learning rate | log-uniform $[10^{-4}, 5 \times 10^{-2}]$ |
| | Outer learning rate | log-uniform $[10^{-5}, 5 \times 10^{-3}]$ |
| | Num. inner steps | $\{1, 2, 4, 8\}$ |
| | Tasks per meta-batch | $\{2, 4, 8\}$ |
| | Num. clusters | $\{2, 3, 4\}$ |
| DKT | Outer learning rate | log-uniform $[10^{-5}, 10^{-2}]$ |
| | GP optimization steps | $\{25, 50, 100, 200\}$ |
| Global Prior | Inner learning rate | log-uniform $[3 \times 10^{-5}, 3 \times 10^{-3}]$ |
| | Inner temperature | log-uniform $[10^{-4}, 10^{-2}]$ |
| | Global prior std. $\sigma$ | log-uniform $[0.02, 0.2]$ |
| | Prior scaling | log-uniform $[10^{3}, 3 \times 10^{4}]$ |
| | Model init. log std. | $\{-4, -3, -2\}$ |
| | Num. inner updates | $\{2, 4, 6\}$ |
| | Outer learning rate | log-uniform $[3 \times 10^{-4}, 3 \times 10^{-3}]$ |
| Causal Prior | Inner learning rate | log-uniform $[3 \times 10^{-5}, 3 \times 10^{-3}]$ |
| | Inner temperature | log-uniform $[10^{-4}, 10^{-2}]$ |
| | Global prior std. $\sigma$ | log-uniform $[0.02, 0.2]$ |
| | Prior scaling | log-uniform $[10^{3}, 3 \times 10^{4}]$ |
| | Model init. log std. | $\{-4, -3, -2\}$ |
| | Num. inner updates | $\{2, 4, 6\}$ |
| | Outer learning rate | log-uniform $[3 \times 10^{-4}, 3 \times 10^{-3}]$ |
| | W learning rate | log-uniform $[10^{-5}, 3 \times 10^{-3}]$ |
| | W regularizer $\gamma_W$ | log-uniform $[10^{-6}, 10^{-1}]$ |
| | Adaptation scale $\alpha$ | $\{0.1, 0.2, 0.3, 0.5\}$ |

## D.6. Other Numerical Results on the UK Biobank Experiment

In this section, we report additional numerical results for all methods compared in the UKBB experiment, including complementary performance metrics and computational runtime statistics. Table 6 shows the AUPRC and Matthews correlation coefficient (MCC) values across all methods. MCC is computed at a fixed 0.5 decision threshold for all methods. Across all tasks, causal methods consistently outperform no-transfer and standard meta-learning baselines. Notably, DKT consistently yields near-zero MCC, suggesting limited threshold level robustness under extreme class imbalance.

Table 7 reports average wall-clock time and peak memory usage on the UK Biobank experiments. The proposed causal meta-learning approach exhibits comparable computational cost to hierarchical Bayesian meta-learning (HBM). Among the methods using expert-inferred embeddings, the MR-based method shows longer runtimes and slightly higher memory usage, consistent with delayed early stopping, indicating slower convergence.

*Table 6.* Average AUPRC and MCC (SD) on target tasks from the UKBB, over 10 seeds. Our method using causal embeddings (ICP, MR) and an expert (**m**edical expert, **n**oisy expert from data, or **d**ata without noise) perform better than the baselines. CHI2 corresponds to our method using correlations. Best in **bold**.

| Method | AUPRC | | | | MCC | | | |
|---|---|---|---|---|---|---|---|---|
| | J44 | J45 | G45 | I21 | J44 | J45 | G45 | I21 |
| CHI2, **m.** | 0.101 (.013) | 0.029 (.003) | 0.019 (.002) | 0.041 (.004) | 0.143 (.018) | 0.025 (.008) | 0.045 (.005) | 0.081 (.008) |
| ICP, **m.** | **0.126 (.006)** | 0.037 (.002) | **0.021 (.001)** | 0.044 (.003) | 0.170 (.011) | 0.054 (.005) | **0.055 (.005)** | **0.085 (.006)** |
| MR, **m.** | 0.091 (.006) | 0.034 (.001) | 0.020 (.001) | 0.044 (.002) | 0.144 (.007) | 0.045 (.003) | 0.051 (.002) | **0.085 (.005)** |
| CHI2, **n.** | 0.074 (.013) | 0.029 (.002) | 0.017 (.001) | 0.039 (.004) | 0.122 (.020) | 0.030 (.007) | 0.040 (.005) | 0.075 (.008) |
| ICP, **n.** | 0.119 (.009) | 0.036 (.001) | 0.019 (.001) | 0.041 (.002) | **0.176 (.005)** | **0.055 (.004)** | 0.047 (.004) | 0.076 (.004) |
| MR, **n.** | 0.107 (.006) | **0.038 (.002)** | 0.020 (.001) | 0.039 (.002) | 0.164 (.009) | **0.055 (.004)** | 0.046 (.003) | 0.070 (.005) |
| CHI2, **d.** | **0.126 (.009)** | 0.032 (.001) | 0.020 (.001) | 0.043 (.003) | 0.174 (.013) | 0.044 (.005) | 0.053 (.003) | 0.082 (.004) |
| ICP, **d.** | 0.120 (.007) | 0.036 (.002) | **0.021 (.001)** | **0.045 (.003)** | 0.172 (.007) | 0.054 (.003) | 0.052 (.004) | **0.085 (.005)** |
| MR, **d.** | 0.085 (.005) | 0.034 (.002) | 0.019 (.001) | 0.044 (.002) | 0.141 (.008) | 0.046 (.003) | 0.048 (.004) | **0.085 (.004)** |
| No transfer | 0.099 (.009) | 0.031 (.002) | 0.018 (.001) | 0.040 (.002) | 0.155 (.007) | 0.037 (.004) | 0.045 (.004) | 0.084 (.004) |
| MAML | 0.099 (.004) | 0.036 (.002) | 0.020 (.001) | 0.042 (.003) | 0.151 (.005) | 0.051 (.003) | 0.051 (.005) | 0.079 (.002) |
| TSA-MAML | 0.101 (.010) | 0.035 (.001) | **0.021 (.002)** | 0.042 (.003) | 0.151 (.007) | 0.049 (.006) | 0.052 (.006) | 0.079 (.005) |
| DKT | 0.117 (.009) | 0.034 (.002) | 0.019 (.001) | 0.038 (.003) | 0.032 (.013) | 0.000 (.000) | 0.004 (.003) | 0.008 (.008) |
| HBM | 0.102 (.006) | 0.035 (.002) | **0.021 (.001)** | 0.043 (.003) | 0.151 (.006) | 0.048 (.003) | 0.052 (.003) | 0.080 (.003) |

*Table 7.* Computational cost of different methods on the UK Biobank experiments. Reported values correspond to average wall-clock time and peak memory usage per task.

| Method | Elapsed Time (hh:mm:ss) | Peak Memory (GiB) |
|---|---|---|
| CHI2, **m.** | 00:09:21 | 5.3 |
| ICP, **m.** | 00:08:54 | 5.0 |
| MR, **m.** | 00:14:34 | 5.5 |
| CHI2, **n.** | 00:13:09 | 7.5 |
| ICP, **n.** | 00:12:13 | 7.1 |
| MR, **n.** | 00:20:39 | 7.7 |
| CHI2, **d.** | 00:08:11 | 7.1 |
| ICP, **d.** | 00:08:21 | 7.0 |
| MR, **d.** | 00:09:47 | 7.1 |
| No transfer | 00:02:06 | 6.5 |
| MAML | 00:03:33 | 6.5 |
| TSA-MAML | 00:03:03 | 1.7 |
| DKT | 00:03:11 | 6.6 |
| HBM | 00:13:27 | 7.1 |

Table 8 compares the linear and nonlinear prior parameterizations ($\phi_t = \theta + \sigma(W z_t)$, where $\sigma$ represents a single hidden layer of dimension 16 with ReLU activation and a linear layer) across all three embeddings methods. For ICP embeddings, the linear parameterization clearly outperforms the nonlinear variant, which collapses toward the global prior (HBM). The pattern is less consistent for MR and CHI2. This suggest that the benefit of the linear parameterization is particularly beneficial when the embedding space captures meaningful task structure, as is the case for ICP embeddings (see Appendix E.3).

Figure 9 illustrates the effect of expert query budget $B$ on AUROC across the UKBB target tasks, using pairwise preference queries from a real medical expert. For ICP embeddings, the performance is stable or increases with query budget from $B = 5$ to $B = 20$. Notably, even at $B = 5$ queries, ICP outperforms the global prior (HBM) on J44 and J45, demonstrating the effectiveness of the expert elicitation framework even in a low-budget regime. In contrast, MR and CHI2 embeddings do

*Table 8.* AUROC (SD) comparison of linear vs. nonlinear prior parametrization on UKBB target tasks (10 seeds). Linear results use oracle embeddings (same as main text). HBM is included as reference.

| Method | J44 | J45 | G45 | I21 |
|---|---|---|---|---|
| ICP, linear | **0.816** (.006) | **0.622** (.004) | **0.673** (.007) | **0.714** (.007) |
| MR, linear | 0.766 (.011) | 0.602 (.006) | 0.651 (.009) | 0.711 (.008) |
| CHI2, linear | 0.815 (.010) | 0.597 (.007) | 0.671 (.007) | 0.705 (.006) |
| ICP, nonlinear | 0.794 (.008) | 0.606 (.006) | 0.669 (.007) | 0.705 (.008) |
| MR, nonlinear | 0.791 (.011) | 0.606 (.008) | 0.667 (.009) | 0.702 (.008) |
| CHI2, nonlinear | 0.772 (.022) | 0.597 (.009) | 0.660 (.008) | 0.690 (.016) |
| HBM | 0.797 (.007) | 0.609 (.007) | 0.670 (.005) | 0.701 (.007) |

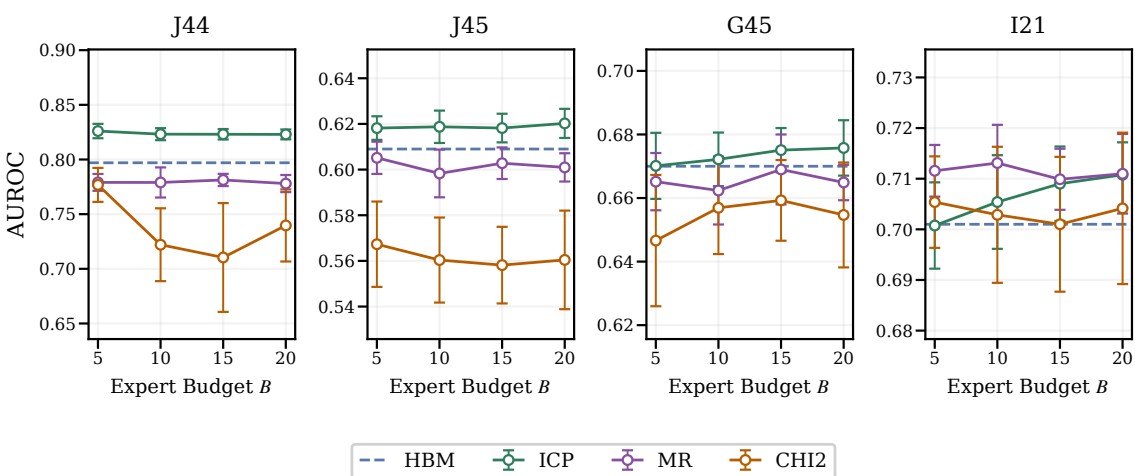

*Figure 9.* AUROC as a function of expert query budget $B$ on UKBB target tasks (10 seeds) with a real medical expert using BALD-based active learning. HBM (dashed) serves as the baseline.

not consistently outperform HBM, and increasing the expert query budget does not lead to systematic improvements. This is consistent with the embedding analysis in Appendix E.3, which shows that MR and CHI2 embedding spaces are less aligned with clinical task similarity, limiting the benefit of expert queries.

# E. Details for the Construction of the Task Embeddings

In this section, we describe the causal and correlational methods used to obtain the task embeddings and analyze their differences. These tasks embeddings are employed in the UK Biobank experiments. We note that the causal inference methods we use employ the longitudinal structure of the dataset. To construct task embeddings we use those disease endpoints $s$ that correspond to other tasks $t$.

### E.1. Causal and Correlational Methods used for Task Embeddings

**Mendelian randomisation.** We estimate causal effects between disease endpoints using two-sample MR implemented using the `TwoSampleMR` R package (Hemani et al., 2017; 2018). We use genome-wide association studies (GWAS) summary statistics from the UK Biobank analyses released by Neale Lab (2018). Genetic instruments for each endpoint are derived from GWAS summary statistics by filtering variants (MAF $> 0.01$, $p < 10^{-5}$) and applying LD clumping (window 10,000kb, $r^2 < 0.01$) using an EUR reference panel and PLINK. For each task $t$, we run MR across exposure endpoints and use the inverse-variance weighted estimate $\beta^{\mathrm{IVW}}$ as the causal effect size. The resulting effect estimates $\beta^{\mathrm{IVW}}_{s,t}$ of the exposures $s = 1, \ldots, K$, on task $t$ are collected into the embedding $z_t = [\beta^{\mathrm{IVW}}_{1,t}, \ldots, \beta^{\mathrm{IVW}}_{K,t}]$.

**Invariant causal prediction.** For each task $t$, we construct a binary outcome indicating whether $t$ occurs in the outcome period and use counts of other tasks observed during exposure years as candidate predictors. We apply invariant causal prediction (ICP; Peters et al., 2016) using the `InvariantCausalPrediction` R package (Meinshausen, 2019) with a random split of the cohort into two environments and a significance level of $\alpha = 0.1$, with boosting-based variable preselection (`maxNoVariables=10`, and `maxNoVariablesSimult=5`). For each task $t$ (endpoint), ICP returns coefficient estimates for the predictors $s = 1, \ldots, K$, whose association with the outcome remains invariant across environments. We form a task embedding $z_t = [\beta_{1,t}, \ldots, \beta_{K,t}]$ by averaging coefficient estimates $\beta$ across accepted sets and setting non-selected coefficients to zero. We use the standard ICP formulation, which assumes no hidden confounders.

**Chi-square ($\chi^2$) temporal association baseline.** As a correlational baseline, we construct task embeddings using chi-square ($\chi^2$) association statistics computed from longitudinal disease co-occurrence data. For each task $t$, we perform a chi-square test between the occurrence of $t$ in an outcome period after the index year and the occurrence of each endpoint $s = 1, \ldots, K$ in the longitudinal data during the exposure period before the index year. The resulting $\chi^2$ statistics are assembled into a vector representation for each task $z_t = [\chi^2_{1,t}, \ldots, \chi^2_{K,t}]$. We compute chi-square statistics using the scikit-learn package (Pedregosa et al., 2011).

### E.2. Uncertainty Quantification of Causal Inference Methods

To assess the reliability of the causal embeddings, we evaluate their stability across different patient subsets using 5-fold cross-validation. For each fold, we hold out 20% of the patients and compute causal embeddings for the remaining 80%. We then measure pairwise cosine similarity between embeddings estimated on different folds, with higher values indicating greater stability. Note that this analysis is performed for ICP and CHI2 only, as MR embeddings are derived from external GWAS summary statistics rather than the patient-level data, and therefore cannot be evaluated in this manner.

Table 9 reports the mean cosine similarity (with standard deviation) across all fold pairs for each source task. CHI2 embeddings are highly stable across all tasks (mean cosine similarity $> 0.9999$). ICP embeddings are stable for most tasks, with lower stability observed for I63 (ischaemic stroke) and J40 (bronchitis). We argue that this instability reflects the genuine causal heterogeneity of the tasks rather than a methodological limitation. J40 groups acute and chronic bronchitis, causally distinct conditions with different mechanisms (viral infection vs. smoking-related airway damage). I63 consists of two distinct pathological pathways: an atherosclerotic pathway and a cardioembolic pathway, each with different upstream causes. Different patient subsets yield different mixtures of these subtypes, causing ICP to correctly identify different causal parents, which is reflected as embedding instability. Notably, these are the only two umbrella codes in our task set grouping multiple causally distinct subtypes and all other tasks show high stability.

*Table 9.* Stability of causal task embeddings estimated using 5-fold cross-validation, measured by mean cosine similarity (standard deviation) across fold pairs.

| Task | ICP | CHI2 |
|------|-----|------|
| E11 | 0.9941 (0.0050) | 0.9999 (0.0001) |
| I10 | 0.9831 (0.0112) | 1.0000 (0.0000) |
| I20 | 0.8889 (0.1177) | 1.0000 (0.0000) |
| I25 | 0.9927 (0.0031) | 0.9990 (0.0001) |
| I48 | 0.9790 (0.0214) | 1.0000 (0.0000) |
| I63 | 0.6133 (0.4629) | 0.9998 (0.0001) |
| J18 | 0.9639 (0.0284) | 0.9998 (0.0002) |
| J40 | 0.6146 (0.3164) | 1.0000 (0.0000) |
| J43 | 0.9739 (0.0105) | 1.0000 (0.0000) |
| **Overall** | **0.8989 (0.2294)** | **0.9999 (0.0001)** |

### E.3. Comparison of the Embeddings

In this section, we compare the task embeddings produced by the different methods considered, and examine how they differ in their resulting task similarity structures.

Figure 10 shows the task embeddings obtained from different causal inference methods, projected onto their first two

principal components (PC1 and PC2) showing the differences in latent embedding space geometry across methods. ICP-derived embeddings exhibit moderately structured geometry with tasks spread along two clinically meaningful axes with cardiovascular-metabolic diseases clustering together and respiratory diseases together. For MR, most of the variance is explained by PC1 (70.9%), making most of the tasks highly concentrated in the embedding space with a few outliers. $\chi^2$-association embeddings have much larger scale (three orders of magnitude), and tasks are widely spread in the embedding space. However, there is a meaningful clustering according to the disease groups. Figure 11 shows the three-nearest neighbor graphs for the causal task embeddings. ICP and CHI2 exhibit similar connective structure with diseases within a same group connected together. MR shows clearly different structure that does not follow the same clinical grouping.

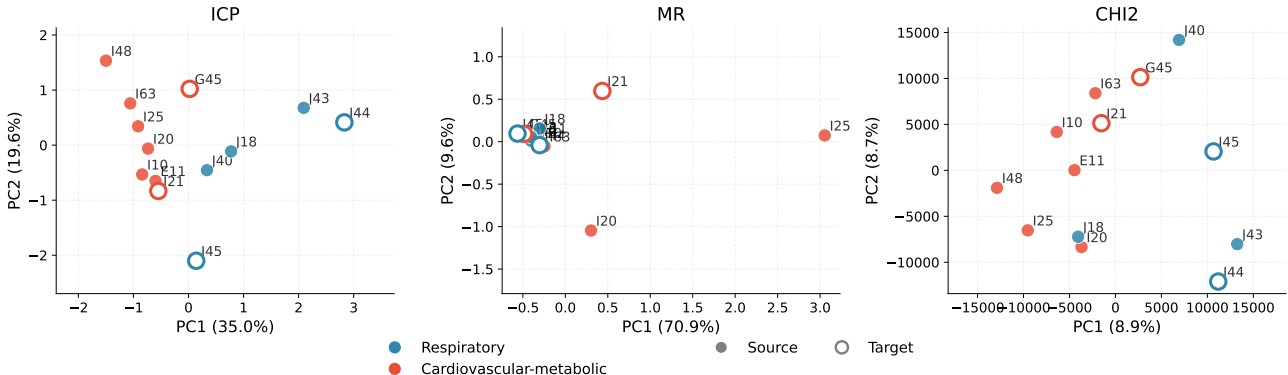

*Figure 10.* Principal component projections of causal task embeddings for Invariant Causal Prediction (ICP), Mendelian Randomisation (MR) and $\chi^2$ (CHI2). Colors indicate disease category (cardiovascular-metabolic vs. respiratory). Hollow markers indicate target tasks.

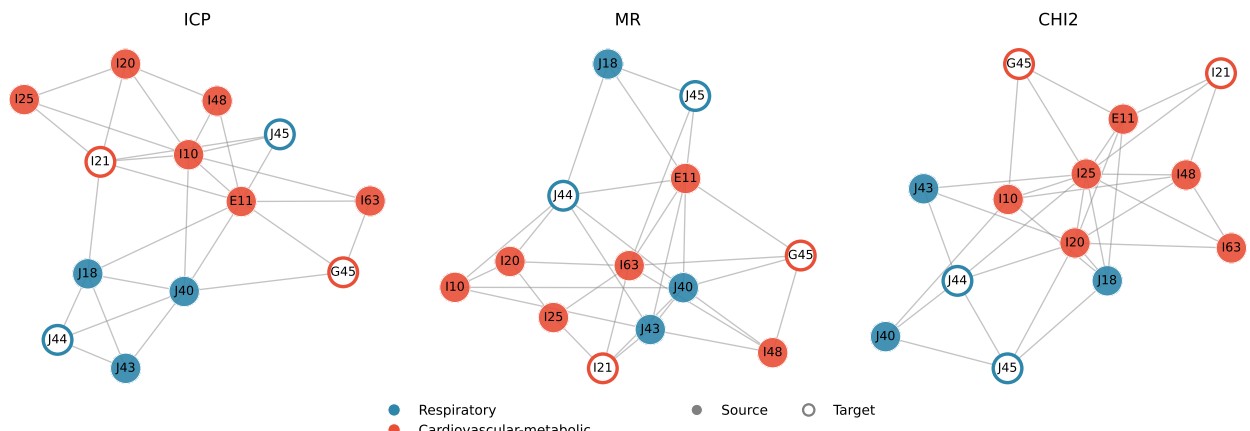

*Figure 11.* Three-nearest neighbor graphs of causal task embeddings. Nodes represent disease prediction tasks and edges connect tasks with similar causal structures. Colors indicate disease category (cardiovascular-metabolic vs. respiratory). Hollow markers indicate target tasks.

The difference of the MR-derived embeddings, with respect to the other two methods, potentially reflects the heterogeneity in strengths of the genetic instruments across the diseases, which is a known challenge in MR (Burgess et al., 2011). Diseases with well-powered GWAS (large sample size and strong genetic association) produce more stable causal estimates, while weaker instruments produce noisier estimated embeddings, which may not reflect the true causal structure.

Table 10 shows the OOD distances for each target task from to the average task ($\bar{z}$) and to the nearest source task. For the ICP method, J44 is the most OOD task with both metrics, and I21 is the least OOD in both metrics as well. MR ranks I21 (myocardial infarction, commonly known as heart attacks) as the most OOD task, which is clinically counterintuitive, since several tasks in the source are known risk factors, for instance type 2 diabetes (E11), and hypertension (I10). This reinforces that MR embeddings is capturing something that does not correspond to clinical similarity, and instead reflects the instrument strength. For CHI2, J45 is the most OOD task, but the distances are very similar throughout.

*Table 10.* Characterization of the target tasks in the embedding space, reported as the Euclidean distance and rank (in parenthesis, 1 indicates the most OOD, 4 is the least) to average source task ($\bar{z}$) and to the nearest source task. CHI2 distances are reported in units of $\times 10^3$. Most OOD task for each metric and method in **bold.**

| | ICP | | MR | | CHI2 | |
|---|---|---|---|---|---|---|
| **Target** | Avg. source | Nearest source | Avg. source | Nearest source | Avg. source | Nearest source |
| J44 | **3.39 (1)** | **2.29 (1)** | 0.69 (4) | 0.59 (3) | 29.19 (4) | 38.63 (4) |
| J45 | 2.61 (2) | 2.10 (3) | 0.94 (2) | 0.72 (2) | **29.54 (1)** | **39.39 (1)** |
| G45 | 1.93 (3) | 2.15 (2) | 0.79 (3) | 0.50 (4) | 29.46 (2) | 39.24 (2) |
| I21 | 1.42 (4) | 1.38 (4) | **1.02 (1)** | **1.23 (1)** | 29.34 (3) | 39.12 (3) |

### E.4. Empirical Validation of Assumption 3 on UK Biobank Data

Assumption 3 states that tasks with similar causal mechanisms have similar optimal predictors. We provide evidence for this on the UK Biobank data. For each pair of tasks $(t, t')$ from all 13 tasks ($\binom{13}{2} = 78$ pairs), we measure the causal embedding distance $\|z_t - z_{t'}\|$ using ICP embeddings, and the predictor similarity. Let $\mathcal{I}_{t,t'} = \{i : x_i \in \mathcal{D}_t \cap \mathcal{D}_{t'}\}$ denote the set of patients shared between the two task cohorts (57k–64k patients per pair). We compare the posterior predictive means $\hat{\phi}_t(x_i) = \mathbb{E}[p(y = 1 \mid x_i, \mathcal{D}_t)]$ from independently trained BNN baselines (no transfer) on these shared patients. To remove the effect of different disease prevalences, we standardize each prediction vector before comparison:

$$d(\hat{\phi}_t, \hat{\phi}_{t'}) = \frac{1}{|\mathcal{I}_{t,t'}|} \sum_{i \in \mathcal{I}_{t,t'}} \left| \frac{\hat{\phi}_t(x_i) - \mu_t}{\sigma_t} - \frac{\hat{\phi}_{t'}(x_i) - \mu_{t'}}{\sigma_{t'}} \right|,$$

where $\mu_t, \sigma_t$ are the mean and standard deviation of $\hat{\phi}_t$ over $\mathcal{I}_{t,t'}$.

Predictions are obtained from independently trained BNN baselines (no transfer). Figure 12 shows the relationship between embedding distance and predictor dissimilarity across all 78 task pairs. Tasks with smaller causal embedding distance yield significantly more similar posterior predictive means (Spearman $r = 0.32$, $p = 0.004$), supporting Assumption 3. The moderate correlation is expected given the real-world confounders and disease heterogeneity.

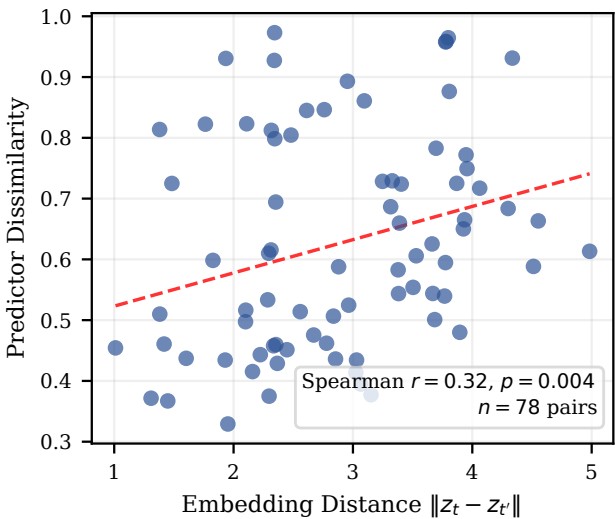

*Figure 12.* Empirical validation of Assumption 3 on UKBB data. Standardized predictor dissimilarity vs. causal embedding distance (ICP) across all 78 tasks pairs. Spearman $r = 0.32$, $p = 0.004$.

