# OpenReview forum: "Bayesian Meta-Learning with Expert Feedback for Task-Shift Adaptation through Causal Embeddings"
_ICML.cc/2026/Conference — ICML 2026 regular_

### Official Review · Reviewer_fp6X · 2026-03-05

**Soundness:** 4
**Presentation:** 3
**Significance:** 4
**Originality:** 3
**Overall Recommendation:** 5
**Confidence:** 4

**Summary:**

This paper studies task-level distribution shift in meta-learning and proposes a causal-aware Bayesian meta-learning framework. The main idea is to learn causal task embeddings for source tasks and use them to parameterize task-specific priors. At deployment, when the target task embedding is unknown, the method queries domain experts with pairwise similarity questions and uses a BALD-based strategy to infer its position in the embedding space.
The paper also provides theoretical analysis on the bias of the learned prior and gives conditions under which the causal prior can outperform a global prior. Experiments on synthetic data and UK Biobank cross-disease prediction tasks are used to evaluate the approach.

**Compliance With Llm Reviewing Policy:**

Affirmed.

**Final Justification:**

The rebuttal addressed your main concerns, raised to 5.

**Key Questions For Authors:**

Assumption 8 assumes a linear relationship between the causal embedding and the optimal task parameters. Is there any empirical evidence that this roughly holds in the UK Biobank experiments?
In Table 1, ICP exp slightly outperforms ICP oracle on J44. Is this just random variation, or could the simulated expert noise have some regularization effect?
The method relies on experts providing pairwise causal similarity judgments. Have the authors considered validating this with real clinicians?

**Limitations:**

yes

**Strengths And Weaknesses:**

Strengths
1. The problem is well motivated. Negative transfer under task distribution shift is a real issue in meta-learning, especially in medical settings. Using causal similarity instead of correlation for transfer is a nice idea and the intuition is easy to understand. The theory section is also fairly clear.
2. The embedding analysis is interesting. The comparison between ICP, MR, and CHI2 gives some useful insight, and the discussion of cases where MR fails to recover clinical groupings makes the evaluation feel more honest.
Weaknesses
1. The theoretical guarantee depends on a fairly strong assumption about a linear relationship between the task embedding and the optimal parameters. It’s not clear how realistic this is, and the paper doesn’t really check whether this assumption holds in the experiments.
2. The experiments raise some questions. The “expert” is simulated using the same methods as the embedding construction, which feels a bit circular. The evaluation also only uses four target tasks, and the improvements are sometimes small or inconsistent.

---

> ### Author Rebuttal · Authors · 2026-03-31
>
> Thank you for your thoughtful questions, and for emphasizing the novelty and intuition of our ideas to approach *real issues* in transfer learning. We are delighted that the comparison between correlational (CHI2) and causal methods (ICP and MR) provides a *useful insight*; that is precisely what we were attempting to do by including the CHI2 comparison. Below we address the weaknesses (W) and key questions (K):
>
> W3 and K1. We truly appreciate that you took the time to go over the proof of our main theorem. Assumption 8 should remind the reader of the LoRA approach, where a low-rank matrix is assumed to be incorporated for a new task. Instead of assuming a specific structure of the adaptation weights, for each task, we assume the simplest relationship with the available information: a line with the causal embeddings. Demonstrating that this assumption is satisfied would be plausible, in theory, if we were able to avoid multimodality and symmetries in the neural-network loss landscape. We still consider this to be a mild assumption; for instance, we are not assuming that $W$ can be easily decomposed, which is the case in standard low-rank methods. Additionally, standard meta-learning approaches assume that there is no such $W$ matrix for different tasks and instead (implicitly) assume that tasks that are causally related are i.i.d. We will explicitly mention these connections in the camera-ready version.
>
>
> W4 and K3. We note that the expert we simulate is imperfect (and not just an oracle) based on the inferred embeddings, and shows performance with what we expect to be the *minimum* of the current medical knowledge. We now additionally involve an expert in results shown in Table 1 below. We see that the performance of our method with ICP embeddings with a medical expert is comparable to an oracle, and is doing better than the simulated expert.
>
>
> K2. We expect the improved performance of the simulated expert versus the oracle to be due to random variation of the simulating mechanism of the expert. We note that the medical expert performs better than the simulated expert, and in some cases the oracle.
>
> Given the importance of the issue our paper addresses, as stated in your original review, and the additional results (including a medical expert!) we have provided to all reviewers, we politely ask that you consider increasing your score to 5.
>
> **Table 1.** Evaluation of AUROC in prediction tasks with a real medical expert for causal meta-learning with ICP embeddings.
>
>
> |            | J44 | J45 | G45 | I21|
> | :--------- | :---------------: | :--------------------: |  :--------------------: |  :--------------------: |
> | ICP, medical expert |     0.823 (.005)| 0.620 (.006)| 0.676 (.009)| 0.711 (.006)|
> | ICP, oracle  | 0.816 (.006)| 0.622 (.004)| 0.673 (.007)| 0.714 (.007)|
> | ICP, simulated expert | 0.820 (.007)| 0.620 (.007)| 0.655 (.007)| 0.693 (.008)|

---

> > ### Author Rebuttal · Reviewer_fp6X · 2026-04-01
> >
> > Thank you for the rebuttal. The real medical expert results in Table 1 directly address my primary concern (W4/K3): the expert achieves performance comparable to or exceeding the oracle across all target tasks, substantially strengthening the practical validity of the method.
> > The LoRA analogy for Assumption 8 is reasonable, though I would encourage a brief empirical diagnostic in the camera-ready version. The explanation for the ICP exp vs. oracle discrepancy (K2) is plausible.
> > In light of the new expert experiment, I am raising my score from 4 to 5.

---

### Official Review · Reviewer_63L5 · 2026-03-10

**Soundness:** 3
**Presentation:** 3
**Significance:** 2
**Originality:** 2
**Overall Recommendation:** 4
**Confidence:** 3

**Summary:**

This paper studies the problem of transfer learning between different tasks. The authors adopt an amortized hierarchical Bayesian meta-learning framework. To model the different causal structures of different tasks, they first extend the framework by conditioning the task-specific prior on causal embeddings. These embeddings of the source task can be obtained through causal inference algorithms. To obtain embeddings of the target task, for which they assume no data is available, the authors propose allowing experts to perform pairwise comparisons, followed by Bayesian preference learning to interpolate the target task's embeddings from the source task's embeddings. The effectiveness of the proposed method is evaluated on synthetic and real-world datasets.

**Compliance With Llm Reviewing Policy:**

Affirmed.

**Final Justification:**

Weak accept. My concerns have been addressed.

**Key Questions For Authors:**

1. Can you provide some examples where we have no data for the target task?

2. How can you evaluate the causal embeddings of the source task (obtained through causal inference) and the target task (obtained using your preference learning algorithm)?

3. Can you include real domain experts in the UK Biobank experiments?

**Limitations:**

No, a potential limitation is the lack of real expert evaluation.

**Strengths And Weaknesses:**

**Strengths**

1. The problem of transfer learning is important.

2. The idea of measuring the similarity between the causal structures of different tasks for transfer learning is intriguing. The authors integrate this concept into a hierarchical Bayesian learning framework, providing a feasible implementation plan.

3. The paper is well-founded, with theoretical justification. The presentation is also clear.

**Weaknesses**

1. I am unsure whether the setting considered in this paper is reasonable. The authors assume that there is no availability of target task data. However, in most transfer learning tasks, such as the UK Biobank data used in their experiments, target task data can be obtained. Therefore, I question the significance of their method.

2. There is a lack of evaluation. The proposed method critically relies on the quality of the causal embeddings (both for the source task and the target task). Thus, evaluating these embeddings is essential. However, the authors did not take this into account in their experiments. More seriously, they did not involve real domain experts to evaluate the feasibility of preference learning, relying instead solely on synthetic expert data. Given the limited improvements against baseline methods (less than or equal to 3 percent), it is difficult to assess the effectiveness of their approach.

---

> ### Author Rebuttal · Authors · 2026-03-31
>
> Thank you for the review! We appreciate your perspective that our paper is *well-founded*, and that our *presentation is clear*. Additionally, we are glad that you found importance in the problem and our approach *intriguing*. We address the specific weaknesses (W) and key questions (K) below:
>
> W1 and K1. We use target tasks where we *do* have access to the outcomes (i.e., diagnoses), since we need to validate the method against known outcomes. That said, the concern for *when* this would be practically applicable is legitimate. Specifically, we expect that our approach could be leveraged in cases with rare outcomes (e.g., rare diseases), or more generally when translating predictors to a new environment -- not only in healthcare. In medicine, a particularly important case is new diseases where knowledge of biological mechanisms can be well-understood by the experts, but due to lack of recorded outcomes, using a machine learning system to diagnose these is hard. To support this claim, we remark that 1) the target tasks have a moderately low prevalence (1-2\%, Table 4 in the appendix), and 2) the main results (Table 1 of the main text) shows how including the expert provides an improved performance over the *No-transfer* baseline.
>
> W2 and K2. (Reliability of embeddings) We agree that evaluating the quality of the causal embeddings is necessary, which is why we included results of this nature in Appendices C.1-2 (synthetic case) and E.2 (UKBiobank). In Appendix E.2, we analyze the estimated embeddings for all three methods in both target and source tasks and compare them against known clinical disease groups. Additionally, in Table 1 below we provide uncertainty quantification (via cosine similarity) of the embeddings in ICP and CHI2 methods, in several folds to illustrate the robustness of the methods. We will include a figure showing this robustness in the camera-ready version. CHI2 is robust across folds. ICP is robust for most of the tasks, showing large standard deviation only for tasks J40 and I63. This instability reflects the actual causal heterogeneity for these disease tasks rather than a methodological limitation. J40 (bronchitis) consists of acute and chronic cases which are causally distinct conditions (viral infections vs. smoking-related airway damage). Similarly, I63 (ischemic stroke) consists of two distinct biological pathways: an atherosclerotic pathway and a cardioembolic pathway. Notably, these two are the only causally heterogenerous tasks in our setting aggregating multiple distinct subtypes; for the remaining (more homogeneous) tasks, the method is consistently stable. Furthermore, we emphasize that the specific causal discovery methods are not our core contribution and our framework is agnostic to the embedding method. As a remark, the MR embeddings we use are based on the work of Neale Lab (2018) where they use instrumental variables and genome-wide association studies (GWAS), so we cannot perform a similar comparison.
>
> W2 and K3. (Incorporating an expert) Good point, Table 2 below shows results including a medical expert for our approach. We see that the performance of our method with ICP embeddings with a medical expert is comparable to an oracle, and is doing better than the original simulated expert.
>
> Thank you for the clear questions, which we believe we have fully addressed, and we find have improved our paper. Given our answers and the additional results we have provided (including the addition of a clinical expert), we hope that our response convinces you to update the score.
>
> **Table 1.** Uncertainty quantification of causal embeddings estimated using 5 folds, and measured by cosine similarity.
>
> | Task | ICP | CHI2 |
> |:-----|:----:|:----:|
> | E11  | 0.9941 (0.0050) | 0.9999 (0.0001) |
> | I10  | 0.9831 (0.0112) | 1.0000 (0.0000) |
> | I20  | 0.8889 (0.1177) | 1.0000 (0.0000) |
> | I25  | 0.9927 (0.0031) | 0.9990 (0.0001) |
> | I48  | 0.9790 (0.0214) | 1.0000 (0.0000) |
> | I63  | 0.6133 (0.4629) | 0.9998 (0.0001) |
> | J18  | 0.9639 (0.0284) | 0.9998 (0.0002) |
> | J40  | 0.6146 (0.3164) | 1.0000 (0.0000) |
> | J43  | 0.9739 (0.0105) | 1.0000 (0.0000) |
> | **Overall** | **0.8989 (0.2294)** | **0.9999 (0.0001)** |
>
> **Table 2.** Evaluation of AUROC in prediction tasks with a real medical expert for causal meta-learning with ICP embeddings.
>
> |            | J44 | J45 | G45 | I21|
> | :--------- | :---------------: | :--------------------: |  :--------------------: |  :--------------------: |
> | ICP, medical expert |     0.823 (.005)| 0.620 (.006)| 0.676 (.009)| 0.711 (.006)|
> | ICP, oracle  | 0.816 (.006)| 0.622 (.004)| 0.673 (.007)| 0.714 (.007)|
> | ICP, simulated expert | 0.820 (.007)| 0.620 (.007)| 0.655 (.007)| 0.693 (.008)|

---

> > ### Author Rebuttal · Reviewer_63L5 · 2026-04-01
> >
> > My concerns have been adequately addressed; therefore, I have raised my score to 4. I encourage the authors to conduct experiments with more medical experts and report the results in their revised version as a more convincing result.

---

### Official Review · Reviewer_ocWh · 2026-03-13

**Soundness:** 3
**Presentation:** 3
**Significance:** 2
**Originality:** 3
**Overall Recommendation:** 4
**Confidence:** 2

**Summary:**

The paper proposes a meta learning technique with Bayesian statistics and causal embeddings that models causal relationships. The authors argue that it is challenging to reason about causal relationships in problem spaces such as healthcare. Specifically, a causally-aware Bayesian meta-learning method is used that conditions task-specific priors on a causal embedding space. The authors provided a theoretical analysis of the negative transfer under distributional shift, and performed an experiment on a simulated dataset as well as the public UK Biobank dataset. Results show improvements against MAML and DKT, etc.

**Compliance With Llm Reviewing Policy:**

Affirmed.

**Key Questions For Authors:**

If authors could present another public dataset benchmark, or another ML task to validate your method, what would that be?

If someone is interested in scaling up the method, which datasets or application areas would you suggest?

**Limitations:**

yes.

**Strengths And Weaknesses:**

Strengths
- The paper shows innovative causal methods for meta-learning. The causal structure model is novel and improves an important baseline: meta-learning
- Experiment results show clear improvements with convincing and clear analysis
- The paper is well written, the theoretical analysis is well presented, the motivation is, and the arguments for the importance of the problem are quite clear

Weaknesses
- The experiment presentation in Table 1 could be clearer. It would help to have a description of the different methods & abbreviations in the table caption, or have a sectionalized or ordered presentation in the table
- Discussions, future work, or experiments on other types of machine learning techniques, e.g., reinforcement learning, will make the results and claim stronger

---

> ### Author Rebuttal · Authors · 2026-03-31
>
> Thank you for the review, and especially for finding our paper overall novel and clear. We will address the specific weaknesses (W) and key questions (K) you indicated, which we note have strengthened our paper.
>
> W1. Thank you for bringing this to our attention! We will cleanup the results on Table 1 of the main text. Further, we will highlight that the *chi-squared* method (labeled CHI2) is non-causal.
>
> W2. Although we have motivated our contributions from a healthcare setting, our contributions are setting-agnostic and can be applied to other transfer learning tasks where an expert can compare the tasks. An intuitive example is in robotics where a robotic arm learns different tasks (e.g., 'open a door', 'open a drawer', etc.), and needs to perform a new task (e.g., 'open a closet'), which can be compared to the original tasks. We will elaborate about other settings and extensions in the Discussion section of the camera-ready version.
>
> K1 and K2. We expect that our method could be used in other areas where theoretical or empirical knowledge of the underlying mechanisms are available, but have not been *easily* embedded into predictions of **new** tasks. Although we are not experts in other areas of knowledge, it seems *plausible* to apply our method in other settings where there is knowledge of the data generating process. A specific example is mentioned in W2 above.
>
> Thank you once again for insightful concerns and highlighting the strengths of our approach. We hope clearing these up would help increase your confidence and score of our paper.

---

> > ### Author Rebuttal · Reviewer_ocWh · 2026-04-04
> >
> > Thanks, authors, for the rebuttal comment. My concerns are fully resolved.

---

### Official Review · Reviewer_pQgk · 2026-03-16

**Soundness:** 3
**Presentation:** 3
**Significance:** 2
**Originality:** 3
**Overall Recommendation:** 4
**Confidence:** 3

**Summary:**

The authors propose a causally-aware Bayesian meta-learning framework for improving adaptation to OOD tasks.
The key idea involves conditioning task-specific priors on latent causal task embeddings, so that transfer is driven by mechanistic similarity over spurious correlations. The approach combines three components (I) causal embeddings, (II) Bayesian meta-learning with priors conditioned on these embeddings, and (3) a preference-learning model that uses pairwise expert comparisons.

The paper also provides some theoretical results linking embedding space similarity to transfer risk. Results on both synthetic and real-world data are reported.

**Compliance With Llm Reviewing Policy:**

Affirmed.

**Final Justification:**

The rebuttal addressed my main concerns and thus changed my evaluation.

**Key Questions For Authors:**

- Could the authors compare their method with meta-learning approaches explicitly designed for related failure modes their algorithm is targetting; task distribution shift (e.g., task-similarity-aware MAML and related methods)? Even if the deployment assumptions are not identifcal, I believe a stronger empirical positioning is warranted.
- The task-specific prior is parametrized as a linear function of the embedding, motivated by favorable theoretical properties. Did you check how nonlinear mappings affect performance?
- Do you have any empirical evidence for the validity of assumption 3, which is central to the method?
- How would the method behave when not all causal mechanisms (target task) are represented among the source tasks? Would we still end up with meaningful priors?

**Limitations:**

Yes.

**Strengths And Weaknesses:**

**Strengths.**
- the problem is well-motivated.
- using causal structure to guide task similarity is appealing
- the paper provides theoretical analysis linking distances in the embedding space to bounds on prior-induced risk.
- both real-world and synthetic experiments are reported.
- expert-guidance is underexplored in the literature, and this method facilitates non-perfect experts, which is almost always the case.

**Weaknesses**
- The comparison with OOD-aware meta-learning methods is limited. Only standard meta-learning algorithms are considered for the i.i.d. task setting (MAML, DKT, hierarchical BMT). This makes it difficult to assess the actual empirical impact of the contribution relative to the SOTA.
- The framework's applicability is relatively limited. In many domains causal structure is difficult to estimate and expert comparisons are unavailable.
- The theoretical guarantees rely heavily on assumption 3 (in prop. 4 and Theorem 5). This assumption is difficult to verify in practice, and may be a stretch.
- The empirical results seem to be quite sensitive to the embedding method used.

---

> ### Author Rebuttal · Authors · 2026-03-31
>
> Thank you for insightful questions and highlighting the key strengths of our approach. Specifically, we are glad you find our problem *well motivated* and that including the (non-perfect!) expert into solving this problem is appealing.
>
> We respond to the weaknesses (W) and key questions (K) you indicated below.
>
> W1 and K1. We agree that task-similarity approaches are relevant as a baseline, and we will include results with task-similarity aware MAML (TSA-MAML; Zhou, et al. 2021) in the same context as Sec. 7.2 (UKBiobank). Table 1 below shows that TSA-MAML performs comparably to standard MAML. This is likely due to the method being unable to find meaningful clusters based on the parameter-space similarity with a limited number of source tasks ($n=9$). In contrast, our method (Linear ICP), performs better by using causal relationships. We expect other correlation based meta-learning approaches to perform similarly in this setting.
>
> W2. Yes, causal structure might not always be readily-available, however we draw the opposite conclusion. Our results show that including causal structures is reasonably easy to do and it **improves** the prediction, this is an encouraging result that will direct applied researchers to incorporate their knowledge into models and embed mechanistic --- not correlational --- structures into predictions. There are plenty of causal discovery approaches which can be applied to learn the structure (up to an error $\varepsilon_{causal}$) and including an expert into the loop is a growing research area (see 'Expert knowledge in learning systems' in Sec. 3 of our main text).
>
> W3 and K3. Regarding your concern about Assumption 3, we remark that it is merely a continuity assumption of the predictions based on the (latent) causal structure. However, the concern about its validity in our case-study is reasonable. We evaluated the (Spearman) correlation between the distances of the no-transfer baselines (Sec. 7.2 of the main text) vs. the distances between the causal embeddings, and obtain a value of $r=0.32$ (with $p=0.004$), supporting our hypothesis that Assumption 3 is valid.
>
> W4. The causal methods we used are Mendelian randomization (MR) and invariant causal prediction (ICP). The other method we used (CHI2), is **non-causal** and only uses correlational information, and performs worse than the causal methods. The sensitivity to the causal discovery method is noted, and we make it explicit in our theoretical derivations (i.e., Prop. 4 and Th. 5). We will modify the caption of the table to ensure that this difference between **causal** and **correlational** embeddings is clear to the reader.
>
> K2. Table 1 below shows a comparison in the UKBiobank with a nonlinearity in the weights with respect to the causal embeddings (i.e., $\phi_t = \theta +\sigma(Wz_t)$). The linear parameterization outperforms the nonlinear mapping, showing that the nonlinear prior collapses towards the global prior. In the synthetic setting, the performance of the nonlinear case (avg. AUROC 0.86) is between our method with linear parameterization (0.90) and the global prior case (0.80).
>
> K4. We suspect that the failure would be case-specific and depend on the reliance of the target task on the missing mechanisms. That said, current methods are not able to incorporate **any** causal information, while our method would include the present mechanisms. In terms of Th. 5, this means that $\varepsilon_{causal}$ is bigger, so the performance will degrade. We will include an analysis with partial causal information in the camera-ready version. The current results show that including only 50\% of the causal parents still outperforms the no causal baseline across all OOD levels. At the highest OOD level ($\varepsilon_{OOD}=4.0$), this corresponds to an AUROC of 0.78, compared to 0.83 (full information) and 0.67 (no causal information).
>
> **Table 1** Average AUROC (SD) with new methods (TSA-MAML), comparison of non-linear and linear version, and other baselines, in target tasks and splits of Sec. 7.2 of the main text.
>
>
> |                    |       J44        |       J45        |       G45        |       I21        |
> | :----------------- | :--------------: | :--------------: | :--------------: | :--------------: |
> | **TSA-MAML**       |   0.792 (.013)   |   0.609 (.010)   |   0.668 (.013)   |   0.701 (.010)   |
> | **MAML**           |   0.791 (.008)   |   0.612 (.006)   |   0.667 (.012)   |   0.701 (.006)   |
> | **Non-linear ICP** |   0.794 (.008)   |   0.606 (.006)   |   0.669 (.007)   |   0.705 (.008)   |
> | **Linear ICP**     | **0.816** (.006) | **0.622** (.004) | **0.673** (.007) | **0.714** (.007) |
> | **HBM**            |   0.797 (.007)   |   0.609 (.007)   |   0.670 (.005)   |   0.701 (.007)   |
>
> Once again, thank you for highlighting the key strengths of our paper. In light of our new experiments, which have improved our paper, and the clarifications, we ask that you consider increasing the score of our paper.

---

> > ### Author Rebuttal · Reviewer_pQgk · 2026-04-03
> >
> > Thank you for the clear rebuttal. My questions are addressed. I will adjust my score accordingly.

---

### Decision · Program_Chairs · 2026-04-30

**Decision:**

Accept (regular)

**Comment:**

This paper proposes a causally-aware Bayesian meta-learning framework for task-shift adaptation, where task-specific priors are conditioned on causal task embeddings and expert pairwise feedback is used when target-task data is limited. Overall, the reviewers found the paper interesting, technically solid, and reasonably novel. The main positives were the importance of the problem, the causal framing of task similarity, and the combination of theory with synthetic and UK Biobank experiments.

The main concerns were limited comparison to stronger baselines, some strong theoretical assumptions, and whether the expert-based setting is realistic enough in practice. I think the rebuttal addressed these points fairly well, especially by adding stronger baseline comparisons and real medical expert results. After rebuttal, all reviewers said their main concerns were resolved, with final recommendations ending up in the weak-accept to accept range. Based on that, my recommendation is weak accept. For the camera-ready version, I would mainly ask for a clearer discussion of the method’s practical scope and limitations, a more careful framing of the theoretical assumptions, and slightly clearer presentation of the main experimental table.